# Genetic loci associated with heart rate variability and their effects on cardiac disease risk

Ilja M. Nolte et al.[#]

Reduced cardiac vagal control reflected in low heart rate variability (HRV) is associated with greater risks for cardiac morbidity and mortality. In two-stage meta-analyses of genome-wide association studies for three HRV traits in up to 53,174 individuals of European ancestry, we detect 17 genome-wide significant SNPs in eight loci. HRV SNPs tag non-synonymous SNPs (in *NDUFA11* and *KIAA1755*), expression quantitative trait loci (eQTLs) (influencing *GNG11*, *RGS6* and *NEO1*), or are located in genes preferentially expressed in the sinoatrial node (*GNG11*, *RGS6* and *HCN4*). Genetic risk scores account for 0.9 to 2.6% of the HRV variance. Significant genetic correlation is found for HRV with heart rate ($-0.74 < r_g < -0.55$) and blood pressure ($-0.35 < r_g < -0.20$). These findings provide clinically relevant biological insight into heritable variation in vagal heart rhythm regulation, with a key role for genetic variants (*GNG11*, *RGS6*) that influence G-protein heterotrimer action in GIRK-channel induced pacemaker membrane hyperpolarization.

#A full list of authors and their affiliations appears at the end of the paper.

Heart rate variability (HRV) is a physiological variation in cardiac cycle duration. When measured under supine or sitting conditions, resting HRV is most prominently centred around the frequency of respiration ($\sim$0.25 Hz) and the intrinsic blood pressure rhythm ($\sim$0.1 Hz). This reflects modulation of tonic activity in the cardiac vagal nerves originating in cortical and subcortical nuclei[1] by oscillatory input at the brainstem level from cardiorespiratory coupling, lung stretch-reflexes, and arterial chemo- and baroreceptors[1,2]. This vagal gating gives rise to oscillatory vagal effects on the pacemaker potentials in the sinoatrial node that scales with the tonic activity in the vagal nerves and provides a source of beat-to-beat variation in heart rate. Due to its good reproducibility[3] and ease of measurement, HRV is a widely used non-invasive research and clinical tool to quantify the degree of vagal control of heart rate[4].

Loss of cardiac vagal control as indexed by low HRV is associated with mortality in patients with cardiovascular disease[5]. Animal research further supports a role for cardiac vagal activity in preventing sudden death and ventricular fibrillation[6]. In addition, hypertension[7], end-stage renal disease[8] and diabetes[9] are all associated with low HRV. Although the above associations may partly reflect impaired cardiac vagal control caused by these diseases, lowered HRV does not simply indicate disease severity as it also predicts all-cause mortality[10] and cardiac morbidity and mortality[11,12] in apparently healthy individuals.

Large inter-individual differences in HRV exist in the basal resting state. Family and twin studies have uniformly confirmed a substantial genetic contribution to resting HRV with heritability estimates between 25 and 71% (ref. 13). Candidate gene studies based on current knowledge of parasympathetic nervous system biology have not yielded results that hold up in replication[14]. To improve our understanding of the genetic basis of HRV, we performed a two-stage meta-analysis of genome-wide association studies (GWAS) in up to 53,174 individuals of European ancestry on three HRV traits (the s.d. of the normal-to-normal inter beat intervals (SDNN), the root mean square of the successive differences of inter beat intervals (RMSSD) and the peak-valley respiratory sinus arrhythmia or high frequency power (pvRSA/HF)). These HRV traits were measured during resting, basal recordings ranging in length from ultrashort 10-s electrocardiograms (ECGs) to up to 90 min of sitting or from 2 to 12 h of daytime recording. Relevance of the identified loci for other ethnicities was examined in data from 11,234 Hispanic/Latino and 6,899 African-American individuals. In silico post-GWAS analyses were performed to test for association with cardiac disease risk factors and disease outcomes and to provide insights into the biological mechanisms by which the identified loci influence cardiac vagal control and its effect on HRV.

We detect 17 SNPs in eight loci harbouring several genes preferentially expressed in the sinoatrial node and significant negative genetic correlations of HRV with heart rate and blood pressure. These findings provide clinically relevant biological insight into heritable variation in vagal heart rhythm regulation, with a key role for genetic variants in proteins (RGS6, GNG11) known to influence G-protein heterotrimer action in GIRK-channel induced pacemaker membrane hyperpolarization.

## Results

### New loci associated with HRV

We meta-analysed results from GWAS on three HRV traits (see Methods section for details) performed by 20 cohorts of European ancestry in up to 28,700 individuals (Fig. 1; Supplementary Figs 1–3; Supplementary Tables 1–4). Using a significance threshold of $1 \times 10^{-6}$,

23 single-nucleotide polymorphism (SNPs) in 14 loci that were associated with one or more of these HRV traits were taken forward for wet-lab genotyping or in silico replication in 11 cohorts including up to 24,474 additional individuals of European ancestry, followed by a second stage meta-analysis (Supplementary Data 1).

After stage 2, we identified 17 lead SNPs (11 independent) in eight loci (Table 1) that reached genome-wide significance ($P < 5 \times 10^{-8}$). The loci on chromosomes 14 and 15 contained three and two independent signals, respectively, (Supplementary Fig. 3). Conditional analysis confirmed the presence of independently associated variants in these loci (Supplementary Table 5). In total, nine independently associated SNPs in seven loci were detected for SDNN, nine independently associated SNPs in eight loci for RMSSD, and five independently associated SNPs in five loci for pvRSA/HF. Many of the SNPs were associated with at least two of the HRV traits (Supplementary Data 1). In four loci, the lead SNPs differed between traits but were in linkage disequilibrium (LD) with each other ($0.24 < r^2 < 0.90$) (Table 1). Forest plots show little heterogeneity in the genetic associations across the entire set of cohorts for all SNPs (Supplementary Fig. 4). Sex-stratified analyses did not show differences in SNP effects between men and women for the genome-wide associated loci (Supplementary Table 6). Separately meta-analysing across cohorts with short laboratory rest recordings versus longer term ambulatory recordings did not suggest sensitivity of the results to these different recording methods (Supplementary Table 7). Results of VEGAS gene-based analyses corroborated those of the SNP-based analyses (Supplementary Note 1).

### Variance explained

Weighted genetic risk scores based on the independent SNPs that reached genome-wide significance after the second stage meta-analysis were computed for the three HRV traits and used to predict RMSSD, SDNN and pvRSA/HF in adults from the Lifelines ($n = 12,101$) and NESDA ($n = 2,218$) cohorts, adolescents from the TRAILS-Pop cohort ($n = 1,191$), and children from the ABCD cohort ($n = 1,094$) (Table 2). The multi-SNP genetic risk scores were all significantly associated with HRV and the percentages of variance explained for the corresponding traits were 1.0–1.4% for SDNN, 1.1–2.4% for RMSSD, and 0.9–2.6% for pvRSA/HF. Cross-trait explained variances of genetic risk scores were close to those for the corresponding trait.

To test the contribution of SNPs that did not reach genome-wide significance, we performed polygenic risk score analyses using increasingly more lenient significance thresholds and determined the percentages of explained HRV in the same four cohorts (Supplementary Fig. 5; Table 3). Maximal variance explained by the polygenic risk score was 0.8–1.4% for SDNN, 0.9–2.3% for RMSSD and 0.9–2.3% for pvRSA/HF. This was reached at relatively small numbers of SNPs ($\leq$71) with additional SNPs adding more noise than signal.

The total variance explained by common SNPs (SNP-based heritability) estimated by Genomic Restricted Maximum Likelihood or LD score regression analysis varied between 10.8 and 13.2%, with only small differences in estimates across methods and HRV traits (Supplementary Note 2).

### Generalization to other ethnicities

In data from up to 11,234 Hispanic/Latino individuals, five SNPs in five of the eight loci identified for RMSSD, seven SNPs in six of the seven loci for SDNN and three SNPs in three of the five loci for pvRSA/HF showed a statistically significant association that was consistent in direction with the association in individuals of European ancestry (Table 4). In data from 6,899 African–Americans, four SNPs from

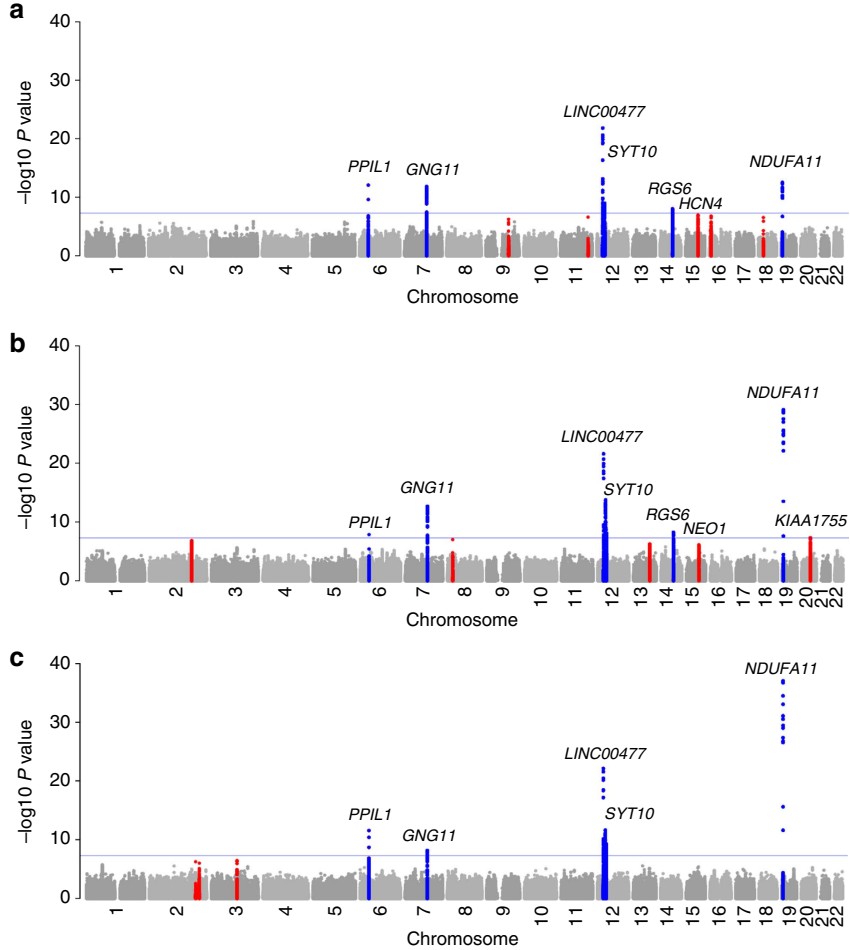

**Figure 1 | Manhattan plots of the meta-analyses of stage 1 GWAS results.** (**a**) SDNN, (**b**) RMSSD and (**c**) pvRSA/HF in up to 28,700 individuals of European ancestry. Only SNPs with a minor allele frequency >1% and that were present in at least 1/3 of the sample are plotted. Significant loci are shown in blue, suggestive ones in red. The blue horizontal line represents the genome-wide significance threshold. Genes closest to the lead SNPs are indicated for the loci that were genome-wide significantly associated with the trait after the stage 1+2 combined meta-analysis.

four of the eight loci were associated with RMSSD, three SNPs in three of the seven loci with SDNN and none with pvRSA/HF. In the combined meta-analysis in a maximum of 71,675 participants from all ethnicities, one SNP (rs6123471 on chromosome 20) was no longer significant (Table 4).

**Correcting HRV for heart rate.** The strong inverse association between HRV and heart rate reflects the well-established simultaneous biological effect of cardiac vagal activity on heart rate[15] and HRV[16], but it also expresses a mathematical dependency of the variance in inter beat interval (IBI) on the mean IBI that is unrelated to the underlying biology. We conducted three analyses to test whether the association of the HRV SNPs was robust to correction of the HRV traits for heart rate (Supplementary Table 8). First, we used a recently developed analytical technique[17] to obtain the meta-analysis for the coefficient of variation of SDNN and RMMSD from the summary statistics of the HRV and resting heart rate meta-analyses[18]. The coefficient of variation detects the amount of IBI variability relative to the mean IBI of each subject, and deals with the proportionality-based dependence of HRV on heart rate[19]. Second, we established the effect of the 17 HRV SNPs on the coefficients of variation for SDNN and RMSSD in the Lifelines, NESDA and TRAILS-Pop cohorts, and meta-analysed the results. Third, we use a mediation analysis in

these same cohorts to see how much of the SNP effects on the three HRV measures was mediated by heart rate. In all three analyses, we find some attenuation of the HRV SNP associations. The average mediation of the association by heart rate was ∼28%. However, the correction for heart rate left most of the HRV SNP associations intact, particularly in the first analysis that used the full discovery sample.

**Association of the HRV SNPs with resting heart rate.** Because the HRV traits reflect cardiac vagal activity, we expected the HRV SNPs to have an effect on resting heart rate. We performed a lookup of the 17 HRV SNPs in a GWAS meta-analysis on resting heart rate in 85,787 individuals[18]. Out of the 17 HRV lead SNPs, 11 were associated with heart rate after correcting for multiple testing (Supplementary Table 9, panel a). All effects were in the expected direction such that the HRV decreasing allele was associated with higher heart rate (Supplementary Fig. 6). Six of the HRV SNPs were not significantly associated with heart rate, including our top hit on chromosome 19 (rs12974991 in *NDUFA11*: $p$ RMSSD = $4.6 \times 10^{-46}$; $p$ heart rate = 0.18). Analysis of summary statistics of the HRV and heart rate meta-analyses as implemented in the *gtx* R package showed that multi-SNP genetic risk scores for HRV were significantly associated with heart rate (Supplementary Table 9, panel b).

**Table 1 | Stage 1 + 2 combined meta-analysis results for SDNN, RMSSD and pvRSA/HF of loci that were genome-wide significant ($P < (5 \times 10^{-8})/3$) in the analysis of individuals of European ancestry.**

| Locus | Chr | SNP | Position (bp) (build 36) | Closest gene | Annotation | Trait | Allele E/O | Stage 1 + 2 | | | |
|---|---|---|---|---|---|---|---|---|---|---|---|
| | | | | | | | | N | EAF | β (s.e.) | P value |
| 1 | 19 | rs12974991* | 5845584 | NDUFA11 | IN | RMSSD | A/G | 43,205 | 0.078 | − 0.116(0.008) | **4.57E-46** |
| | | rs12974440* | 5845386 | | IN | pvRSA/HF† | A/G | 29,527 | 0.073 | − 0.244(0.019) | **1.91E-41** |
| | | rs12980262* | 5844058 | | M | SDNN | A/G | 46,046 | 0.076 | − 0.060(0.006) | **2.30E-23** |
| 2 | 12 | rs10842383 | 24663234 | LINC00477 (C12orf67) | IG, HR[i] | SDNN | C/T | 47,808 | 0.863 | − 0.049(0.004) | **9.33E-31** |
| | | | | | | RMSSD | | 43,223 | 0.862 | − 0.065(0.006) | **2.45E-29** |
| | | | | | | pvRSA/HF† | | 31,085 | 0.865 | − 0.124(0.013) | **1.20E-25** |
| 3 | 6 | rs236349 | 36928543 | PPIL1 | IG | SDNN | G/A | 51,379 | 0.651 | − 0.033(0.003) | **3.70E-25** |
| | | | | | | RMSSD | | 46,795 | 0.655 | − 0.035(0.004) | **9.10E-17** |
| | | | | | | pvRSA/HF† | | 33,654 | 0.645 | − 0.069(0.009) | **3.16E-15** |
| 4 | 12 | rs7980799‡ | 33468257 | SYT10 | IN, HR[ii] | RMSSD | A/C | 44,210 | 0.390 | − 0.039(0.004) | **3.19E-20** |
| | | rs1351682‡ | 33490042 | | IG, HR[iii] | pvRSA/HF† | G/A | 30,643 | 0.437 | − 0.073(0.009) | **5.70E-15** |
| | | rs1384598‡ | 33514166 | | IG, HR[iv] | SDNN | T/A | 47,358 | 0.432 | − 0.023(0.003) | **7.37E-13** |
| 5 | 7 | rs4262§ | 93389364 | GNG11 | UTR5, Q, HR[v] | SDNN | C/T | 49,005 | 0.390 | − 0.028(0.003) | **4.26E-17** |
| | | | | | | pvRSA/HF† | | 31,281 | 0.388 | − 0.050(0.010) | **1.84E-11** |
| | | rs180238§ | 93388383 | | UP, Q, HR[vi] | RMSSD | C/T | 44,420 | 0.333 | − 0.034(0.004) | **7.99E-16** |
| 6 | 14b | rs4899412‖ | 71534015 | RGS6 | IN, Q | SDNN | T/C | 48,252 | 0.253 | − 0.026(0.004) | **3.13E-13** |
| | | rs2052015‖ | 71556806 | | | RMSSD | T/C | 45,492 | 0.165 | − 0.036(0.006) | **3.56E-10** |
| | 14c | rs2529471 | 71883022 | | IN | SDNN | C/A | 49,619 | 0.429 | − 0.021(0.003) | **1.88E-12** |
| | 14a | rs36423 | 71422955 | | IG | SDNN | T/G | 48,182 | 0.129 | − 0.033(0.005) | **6.25E-13** |
| | | | | | | RMSSD | | 45,419 | 0.127 | − 0.040(0.006) | **5.36E-11** |
| 7 | 15a | rs2680344 | 71440538 | HCN4 | IN, HR[vii] | SDNN | A/G | 51,370 | 0.777 | − 0.024(0.004) | **4.88E-11** |
| | 15b | rs1812835 | 71294557 | NEO1 | IN, Q | RMSSD | A/C | 44,421 | 0.418 | − 0.025(0.004) | **5.18E-10** |
| 8 | 20 | rs6123471 | 36273570 | KIAA1755 | UTR3, HR[viii] | RMSSD | T/C | 46,789 | 0.534 | − 0.024(0.004) | **1.30E-08** |

Allele E/O, effect allele/other allele; bp, base pair position based on build 36 (hg18); Chr, chromosome; EAF, effect allele frequency; HR, HRV SNPs that are in pairwise LD (based on SNAP, HapMap release 22 CEU) with identified loci associated with heart rate (HR) from den Hoed et al.[18]; IG, intergenic variant; IN, intronic variant; N, sample size; M, missense variant; Q, associated with an eQTL; s.e., standard error of β; UTR3, variant in the 3′ untranslated region; UTR5, variant in the 5′ untranslated region; UP, upstream variant (within 2kb); β, effect size.
NOTE: Only SNPs that were independently associated (that is, lead SNPs) to the traits are shown. At some loci lead SNPs were the same for the different traits, at other loci there were different (dependent) lead SNPs for the different traits. SNPs are sorted according to P value of the combined meta-analysis per locus. Genome-wide significant association (two-sided $P < 5 \times 10^{-8}$), corrected for testing three traits (that is, $P < 5 \times 10^{-8}/3$), is shown in bold. Effect alleles were chosen to reflect an increased risk for *low* levels of HRV, hence β's are all negative.
[i] $r^2 = 1$ between rs10842383 and rs17287293[HR]; [ii] same SNP; [iii] $r^2 = 0.782$ between rs1351682 and rs7980799[HR]; [iv] $r^2 = 0.695$ between rs1384598 and rs7980799[HR]; [v] $r^2 = 0.570$ between rs426[2] and rs180242[HR]; [vi] $r^2 = 0.893$ between rs180238 and rs180242[HR]; [vii] $r^2 = 0.505$ between rs2680344 and rs4489968[HR]; [viii] $r^2 = 1$ between rs6123471 and rs6127471[HR].
*these SNPs are all in perfect LD ($r^2 = 1$).
†P value, allele, EAF, N from P value weighted meta-analysis of all cohorts using METAL and β, s.e. from inverse-variance meta-analysis of only HF cohorts using GWAMA.
‡$r^2 = 0.782$ between rs7980799 and rs1351682; $r^2 = 0.695$ between rs7980799 and rs1384598; $r^2 = 0.903$ between rs1351682 and rs1384598.
§$r^2 = 0.600$ between rs4262 and rs180238.
‖$r^2 = 0.237$ between rs4899412 and rs2052015.

In addition, genetic risk scores based on the independent genome-wide significant HRV SNPs from the combined stage 1 and 2 meta-analysis were tested for association with heart rate in the Lifelines, NESDA, TRAILS-Pop and ABCD cohorts (Supplementary Table 9, panel b). The three multi-SNP risk scores of the HRV traits explained a small, but mostly significant percentage of variance in heart rate (0.09–1.13%). Polygenic risk score analysis showed that adding HRV SNPs below the genome-wide significance threshold did not further increase the variance explained in heart rate (Supplementary Fig. 5; Supplementary Table 9, panel c).

The reverse question, whether SNPs with effects on heart rate are associated with HRV, was also investigated. The 21 heart rate SNPs identified by the GWAS meta-analysis on heart rate[18] explained between 0.2 and 0.9% of the variance in the three HRV traits (Supplementary Table 10).

**Association with cardiometabolic traits and diseases.** In addition to heart rate, we examined the association of the 17 HRV-associated SNPs with other confirmed risk factors for cardiac, metabolic and renal disease traits and endpoints using data from large-scale GWAS meta-analyses (Supplementary Table 11). Multi-SNP risk scores were computed based on our 17 top SNPs and we tested their association with the outcomes.

No effects of risk scores using the 17 HRV SNPs were observed for systolic or diastolic blood pressure, body mass index, renal function, heart failure, sudden cardiac death, coronary artery disease, atrial fibrillation or type 2 diabetes. Only for atrial fibrillation we observed individually significant SNPs. These two highly significant SNPs (rs10842383 near LINC00477, $P = 3.45 \times 10^{-7}$ and rs2680344 in HCN4, $P = 4.34 \times 10^{-7}$ (Supplementary Table 12) had large opposite effects on atrial fibrillation, while both decreased HRV. In addition to these lookups that were restricted to genome-wide significant SNPs, we employed bivariate LD score regression[20] that uses the full GWAS summary statistics of the HRV and cardiometabolic traits and diseases to compute genetic correlations. The genetic correlations systematically pointed to an overlap in the genetic variants causing low HRV and increased risk for disease (that is, negative

**Table 2 | Explained variance in HRV traits in the Lifelines ($n = 12,101$), NESDA ($n = 2,118$), TRAILS-Pop ($n = 1,191$) and ABCD ($n = 1,094$) cohorts by the weighted multi-SNP genetic risk score based on the independent genome-wide significant SNPs in the stage $1 + 2$ meta-analysis.**

| Trait | Risk score | Lifelines | | | NESDA | | | TRAILS-Pop | | | ABCD | | |
|---|---|---|---|---|---|---|---|---|---|---|---|---|---|
| | | No. SNPs | P value | $\Delta R^2$ | No. SNPs | P value | $\Delta R^2$ | No. SNPs | P value | $\Delta R^2$ | No. SNPs | P value | $\Delta R^2$ |
| SDNN | SDNN | 9 | 6.3E-33 | 1.00% | 9 | 4.9E-10 | 1.39% | 10 | 8.0E-05 | 1.28% | 10 | 7.8E-04 | 1.03% |
| SDNN | RMSSD | 7 | 1.1E-30 | 0.93% | 11 | 7.5E-10 | 1.35% | 11 | 6.3E-06 | 1.69% | 11 | 3.5E-05 | 1.56% |
| SDNN | pvRSA/HF | 5 | 6.4E-25 | 0.75% | 5 | 5.2E-07 | 0.89% | 5 | 8.7E-07 | 1.99% | 4 | 4.3E-03 | 0.75% |
| RMSSD | SDNN | 9 | 8.3E-37 | 1.13% | 9 | 2.0E-11 | 1.54% | 10 | 4.4E-06 | 1.73% | 10 | 4.7E-04 | 1.11% |
| RMSSD | RMSSD | 7 | 8.8E-37 | 1.13% | 11 | 6.1E-12 | 1.62% | 11 | 5.3E-08 | 2.42% | 11 | 2.5E-06 | 2.01% |
| RMSSD | pvRSA/HF | 5 | 1.5E-30 | 0.93% | 5 | 2.0E-11 | 1.54% | 5 | 2.1E-09 | 2.92% | 4 | 8.1E-04 | 1.02% |
| pvRSA/HF | SDNN | 7 | NA | NA | 9 | 1.5E-13 | 1.58% | 10 | 4.3E-05 | 1.38% | 10 | 2.0E-03 | 0.87% |
| pvRSA/HF | RMSSD | 6 | NA | NA | 11 | 7.1E-14 | 1.62% | 11 | 1.3E-06 | 1.93% | 11 | 1.5E-05 | 1.70% |
| pvRSA/HF | pvRSA/HF | 5 | NA | NA | 5 | 7.7E-17 | 2.01% | 5 | 1.4E-08 | 2.64% | 4 | 1.8E-03 | 0.89% |

NA, not available.

NOTE: $\Delta R^2$ is the difference in percentage of explained variance by the multi-SNP genetic or polygenetic risk score between the models with and without the risk score while adjusting both for age, sex and principal components.

For Lifelines, NESDA and TRAILS-Pop the weights (that is, effects sizes) and number of genome-wide significant SNPs included in the risk score were adjusted by analytically extracting the cohort's effect size and s.e. from the meta effect size and s.e., respectively, and recalculating the P value based on these adjusted effect sizes and s.e.'s, since these cohorts were included in stage 1 and/or 2.

correlations with systolic and diastolic blood pressure, coronary artery disease, heart failure, sudden cardiac death, BMI and type 2 diabetes) compatible with clinical relevance of the HRV SNPs identified, although significance was reached only for systolic and diastolic blood pressure after correction for number of outcomes tested (Supplementary Table 11).

**Potential functional impact of the HRV variants**. To identify functional variants tagged by the 17 HRV SNPs, we performed various post-GWAS annotation (Supplementary Fig. 7). *In silico* annotation (Supplementary Data 2) showed that the lead SNP for SDNN on chromosome 19 was a non-synonymous SNP (rs12980262 in *NDUFA11*) and that the lead SNPs for RMSSD (rs12974991) and pvRSA/HF (rs12974440) were in perfect LD with this SNP (Table 1; Supplementary Data 2). SNP rs129080262 was characterized as deleterious, with a sorting intolerant from tolerant (SIFT) score of 0.01 and a polymorphism phenotyping (PolyPhen) score of 0.753 indicating a possibly damaging effect. Functional variant analyses using RegulomeDB confirmed that rs12980262 and rs12974440 in *NDUFA11* on chromosome 19 likely have functional consequences (Supplementary Table 13) by binding to transcription factors or influencing the chromatin state. SNP rs6123471 in the locus on chromosome 20 was in high LD with two non-synonymous SNPs in the *KIAA1755* gene (rs3746471 [$r^2 = 0.94$] and rs760998 [$r^2 = 0.55$]) that are predicted to yield tolerated, benign amino acid changes (Supplementary Data 2).

We examined if the 17 HRV SNPs were eQTLs in a large whole blood database. Four of the HRV SNPs were significantly (false discovery rate $< 5\%$) associated with gene expression in blood (Supplementary Table 14): rs1812835 with expression of *NEO1*, rs4899412 with expression of *RGS6*, and rs180238 and rs4262 with expression of *GNG11*. These four SNPs were all in strong LD with the top eQTL SNPs for these genes ($r^2 > 0.70$) and lost significance after conditioning on the corresponding top eQTL. The eQTLs for *NEO1* and *RGS6* were replicated in at least one other whole blood eQTL study (Supplementary Table 14). The eQTL for *GNG11* was replicated in the medulla ($P = 2.8 \times 10^{-4}$) and the anterior tibialis artery ($P = 8.1 \times 10^{-9}$). None of the 17 SNPs reached significance in a smaller heart eQTL database.

Nine of the 17 HRV SNPs were in high LD ($r^2 > 0.70$) with SNPs associated with methylation level of one or multiple CpG sites (methylation quantitative trait loci [mQTLs]) in whole blood

(Supplementary Table 15). Two of the HRV SNPs that were eQTLs also influenced methylation of the same gene in whole blood, strongly suggestive of a regulatory function for those SNPs. eQTL rs1812835 in *NEO1* was associated with methylation level of cg11357013, cg19281068, cg11552023 and cg17150474. eQTL rs4262 was associated with methylation level of cg08038054 and cg06439941 in *GNG11*. The other two eQTL SNPs did not achieve genome-wide significance level for an association with methylation, but eQTL rs4899412 in *RGS6* was in high LD with a proxy SNP (rs2238280) that was associated with methylation level of cg19493789, which is located in a CpG island shelf near *RGS6*.

Five other HRV SNPs were (in high LD with) mQTLs but were not themselves eQTLs. For example, HRV SNPs rs12974991, rs12974440 and rs12980262 (chromosome 19) were associated with methylation level of multiple CpG sites (cg22854549, cg03715305 and cg19211619) located in or nearby *NDUFA11*, but were not associated with expression level of *NDUFA11* in whole blood. Such mQTLs may well exert a regulatory effect on *NDUFA11* in other tissues. DEPICT tissue enrichment analysis (Supplementary Data 3; Supplementary Table 16, Supplementary Fig. 8) showed *NDUFA11* expression was weak in blood, but enriched in heart, sensory and endocrine tissues.

## Discussion

This meta-analysis of GWAS for HRV yielded 17 lead SNPs (11 independent) in eight loci that were genome-wide significantly associated, six of which generalized to individuals of African-American and Hispanic/Latino ethnicity. Various ways that correct HRV for its mathematical dependency on resting heart rate attenuated the SNP effects, but largely left the associations intact. Together, the hits in the eight loci explained 0.9–2.6% of the variance in resting HRV in four independent cohorts of European ancestry. Details of known biological functions of the genes closest to these loci are given in the Supplementary Note 6.

We noted a strong enrichment of our HRV loci in a previously conducted meta-analysis of GWAS for resting heart rate[18], a known risk factor for cardiac morbidity and mortality[21,22]. SNPs in five of the 21 resting heart rate loci (that is, *LINC00477 (C12orf67)*, *SYT10*, *GNG11*, *HCN4* and *KIAA1755*) were associated with HRV at genome-wide significance level and six more attained nominal significance, with associations always in the expected direction. Genetic risk

**Table 3 | Explained variance in HRV traits in the Lifelines ($n = 12,101$), NESDA ($n = 2,118$), TRAILS-Pop ($n = 1,191$) and ABCD ($n = 1,094$) cohorts by the optimal polygenic risk scores computed at the $P$ value threshold that explained the largest percentage of phenotypic variance.**

| Trait | Risk score | Cohort | P cutoff | No. SNPs | P value | $\Delta R^2$ |
|---|---|---|---|---|---|---|
| SDNN | SDNN | Lifelines | <5E-7 | 13 | 6.8E-27 | 0.82% |
| | | NESDA | <5E-8 | 6 | 2.6E-08 | 1.16% |
| | | TRAILS-Pop | <5E-5 | 64 | 1.1E-04 | 1.23% |
| | | ABCD | <5E-5 | 71 | 9.4E-05 | 1.39% |
| SDNN | RMSSD | Lifelines | <5E-8 | 8 | 2.4E-23 | 0.71% |
| | | NESDA | <5E-6 | 23 | 1.2E-07 | 1.05% |
| | | TRAILS-Pop | <5E-8 | 8 | 1.2E-04 | 1.23% |
| | | ABCD | <5E-7 | 13 | 2.8E-06 | 2.00% |
| SDNN | pvRSA/HF | Lifelines | <5E-8 | 7 | 3.1E-19 | 0.58% |
| | | NESDA | <5E-8 | 4 | 3.5E-05 | 0.64% |
| | | TRAILS-Pop | <5E-7 | 6 | 9.7E-06 | 1.61% |
| | | ABCD | <5E-5 | 67 | 9.2E-04 | 1.01% |
| RMSSD | SDNN | Lifelines | <5E-7 | 13 | 8.9E-31 | 0.95% |
| | | NESDA | <5E-8 | 6 | 1.6E-10 | 1.46% |
| | | TRAILS-Pop | <5E-8 | 7 | 8.3E-06 | 1.63% |
| | | ABCD | <5E-5 | 71 | 1.6E-04 | 1.30% |
| RMSSD | RMSSD | Lifelines | <5E-7 | 12 | 2.8E-30 | 0.94% |
| | | NESDA | <5E-7 | 10 | 2.7E-10 | 1.43% |
| | | TRAILS-Pop | <5E-7 | 11 | 3.4E-07 | 2.13% |
| | | ABCD | <5E-7 | 13 | 3.8E-07 | 2.34% |
| RMSSD | pvRSA/HF | Lifelines | <5E-8 | 7 | 1.4E-25 | 0.78% |
| | | NESDA | <5E-8 | 4 | 3.6E-09 | 1.25% |
| | | TRAILS-Pop | <5E-7 | 6 | 3.7E-08 | 2.47% |
| | | ABCD | <5E-8 | 67 | 8.4E-04 | 1.02% |
| pvRSA/HF | SDNN | NESDA | <5E-8 | 6 | 1.1E-12 | 1.52% |
| | | TRAILS-Pop | <5E-8 | 7 | 5.0E-05 | 1.36% |
| | | ABCD | <5E-5 | 71 | 5.4E-04 | 1.09% |
| pvRSA/HF | RMSSD | NESDA | <5E-7 | 10 | 5.6E-14 | 1.69% |
| | | TRAILS-Pop | <5E-7 | 11 | 3.3E-06 | 1.78% |
| | | ABCD | <5E-7 | 13 | 1.9E-06 | 2.06% |
| pvRSA/HF | pvRSA/HF | NESDA | <5E-8 | 4 | 4.4E-13 | 1.58% |
| | | TRAILS-Pop | <5E-7 | 6 | 1.6E-07 | 2.25% |
| | | ABCD | <5E-5 | 67 | 1.6E-03 | 0.90% |

NA, not available.
NOTE: Weighted polygenic risk score was determined based on independent SNPs in the stage 1 meta-analysis. For NESDA and TRAILS-Pop the weights (that is, effects sizes) and $P$ values were adjusted by analytically extracting the cohort's effect size and s.e. from the meta effect size and s.e., respectively, and recalculating the $P$ value based on these adjusted effect size and s.e., since these cohorts were included in stage 1.

scores for HRV traits were also significantly associated with heart rate and LD score regression confirmed that the allelic variants that decrease HRV in parallel increase heart rate. This suggests to us that part of the HRV SNPs exert their effect on heart rate through oscillatory modulation of pacemaker activity by the vagal nerves.

Supplementary Fig. 9 depicts the two routes by which acetylcholine released by the vagal nerves in the sinoatrial node is known to influence heart rate, both of which are supported by our results in GNG11, RGS6 and HCN4. By binding the muscarinic type 2 receptor ($M_2R$) and dissociating the G-protein heterotrimer ($G\alpha\beta\gamma$) into a $G\alpha_{i/o}$ subunit and a $G\beta\gamma$ component, acetylcholine inhibits the ongoing depolarization of the pacemaker cells by $\beta1/\beta2$-adenylatecyclase activation of funny ($I_f$) channels and calcium channels[23]. In parallel, it acts to actively hyperpolarize the pacemaker cells by activation of the GIRK1/4 channel. Each route accounts for about half of the

tonic decrease in heart rate upon vagal stimulation[23], but the response time for $M_2R$-GIRK effects on the sinus rate is much shorter than for the $M_2R$-HCN2/4 or the $\beta1/\beta2$-adenylatecyclase signalling pathways. Only signalling through the $G\beta\gamma$ component is fast enough ($\sim 0.3$ s) to rapidly track changes in vagal outflow to the sinoatrial node, for example, as they occur within the duration of a single respiration ($\sim 4.5$ s), whereas signalling through the $\alpha$ subunit is too slow ($> 3$ s) to track such phasic changes in acetylcholine release[1,24]. GIRK signalling, therefore, accounts for most of HRV due to the phasic oscillation in vagal activity[24], but it accounts for only half of the tonic vagal effects on heart rate.

The above leads to HRV only partially capturing the vagal effects on heart rate. Additional reasons for the imperfect relation between HRV and vagal effects on heart rate[1,2] are individual differences in: (i) resting respiration rate and depth; (ii) the amplitude of the intrinsic 0.1 Hz oscillations related to both vagal

**Table 4 | Meta-analysis results for the identified loci in other ethnicities and combined meta-analysis results with European ancestry.**

| Locus | Chr | SNP | Trait | Allele E/O | Hispanic/Latino | | | | African American | | | | EUR + HIS + AfAm |
|---|---|---|---|---|---|---|---|---|---|---|---|---|---|
| | | | | | N | EAF | β (s.e.) | P value | N | EAF | β (s.e.) | P value | P value |
| 1 | 19 | rs12974991 | RMSSD | A/G | 11,233 | 0.065 | −0.162 (0.018) | **7.05E-20** | 6,673 | 0.455 | −0.077 (0.033) | **1.10E-02** | **1.86E-63** |
| | | rs12974440 | pvRSA/HF* | A/G | 404 | 0.048 | −0.518 (0.174) | **3.06E-03** | 1900 | 0.019 | −0.189 (0.158) | 3.41E-01 | **4.53E-41** |
| | | rs12980262 | SDNN | A/G | 11,233 | 0.048 | −0.161 (0.070) | **1.04E-02** | 6675 | 0.093 | −0.046 (0.030) | 6.48E-02 | **1.57E-24** |
| 2 | 12 | rs10842383 | SDNN | C/T | 11,233 | 0.854 | −0.053 (0.012) | **2.45E-06** | 6676 | 0.955 | 0.056 (0.026) | 9.83E-01 | **7.61E-33** |
| | | | RMSSD | | 11,233 | 0.854 | −0.064 (0.012) | **1.38E-07** | 6673 | 0.955 | 0.065 (0.030) | 9.86E-01 | **4.23E-32** |
| | | | pvRSA/HF* | | 404 | 0.830 | −0.140 (0.095) | 1.40E-01 | 1901 | 0.959 | 0.068 (0.104) | 6.79E-01 | **4.98E-25** |
| 3 | 6 | rs236349 | SDNN | G/A | 11,234 | 0.684 | −0.034 (0.009) | **6.15E-05** | 6676 | 0.724 | −0.017 (0.011) | 6.57E-02 | **1.76E-28** |
| | | | RMSSD | | 11,234 | 0.684 | −0.034 (0.009) | **1.67E-04** | 6673 | 0.724 | −0.021 (0.013) | **4.79E-02** | **5.88E-20** |
| | | | pvRSA/HF* | | 404 | 0.704 | −0.164 (0.080) | **4.13E-02** | 1901 | 0.729 | 0.004 (0.043) | 4.87E-01 | **4.64E-15** |
| 4 | 12 | rs7980799 | RMSSD | A/C | 11,234 | 0.269 | −0.031 (0.010) | **1.23E-03** | 6488 | 0.097 | −0.029 (0.021) | 7.70E-02 | **1.57E-22** |
| | | rs1351682 | pvRSA/HF* | G/A | 404 | 0.348 | −0.166 (0.077) | **3.19E-02** | 1901 | 0.142 | −0.082 (0.058) | 6.91E-02 | **2.00E-14** |
| | | rs1384598 | SDNN | T/A | 11,234 | 0.307 | −0.026 (0.009) | **1.80E-03** | 6676 | 0.146 | −0.024 (0.015) | 5.41E-02 | **2.88E-15** |
| 5 | 7 | rs4262 | SDNN | C/T | 11,234 | 0.427 | −0.016 (0.008) | **2.39E-02** | 6676 | 0.608 | −0.028 (0.011) | **5.87E-03** | **5.36E-19** |
| | | | pvRSA/HF* | | 404 | 0.410 | −0.014 (0.074) | 8.46E-01 | 1901 | 0.618 | −0.055 (0.043) | 1.15E-01 | **1.50E-11** |
| | | rs180238 | RMSSD | C/T | 11,234 | 0.367 | −0.024 (0.009) | **4.05E-03** | 6673 | 0.474 | −0.032 (0.011) | **2.77E-03** | **8.07E-19** |
| 6 | 14b | rs4899412 | SDNN | T/C | 11,234 | 0.329 | −0.012 (0.009) | 8.27E-02 | 6676 | 0.419 | −0.009 (0.010) | 1.86E-01 | **5.96E-13** |
| | | rs2052015 | RMSSD | T/C | 11,234 | 0.173 | −0.015 (0.012) | 9.94E-02 | 6673 | 0.098 | −0.001 (0.020) | 4.83E-01 | **1.94E-09** |
| | 14c | rs2529471 | SDNN | C/A | 11,233 | 0.485 | −0.018 (0.008) | **1.38E-02** | 6676 | 0.543 | 0.003 (0.010) | 3.83E-01 | **2.08E-12** |
| | 14a | rs36423 | SDNN | T/G | 11,234 | 0.193 | −0.021 (0.011) | **2.41E-02** | 6676 | 0.160 | −0.030 (0.015) | **1.79E-02** | **1.60E-14** |
| | | | RMSSD | | 11,234 | 0.193 | −0.017 (0.011) | 7.05E-02 | 6673 | 0.160 | −0.034 (0.016) | **1.72E-02** | **1.02E-11** |
| 7 | 15a | rs2680344 | SDNN | A/G | 11,234 | 0.681 | −0.005 (0.009) | 2.97E-01 | 6676 | 0.450 | −0.024 (0.011) | **1.32E-02** | **2.90E-11** |
| | 15b | rs1812835 | RMSSD | A/C | 11,234 | 0.426 | −0.012 (0.009) | 8.83E-02 | 1388 | 0.140 | −0.009 (0.033) | 3.98E-01 | **5.30E-10** |
| 8 | 20 | rs6123471 | RMSSD† | T/C | 11,234 | 0.560 | −0.001 (0.009) | 4.40E-01 | 6673 | 0.739 | 0.020 (0.013) | 5.67E-02 | 5.14E-06 |

AfAm, African American; Allele E/O, effect allele/other allele; Chr, chromosome; bp, base pair position based on build 36 (hg18); EAF, effect allele frequency; EUR, European; HIS, Hispanic/Latino; N, sample size; s.e., standard error of β; β, beta/effect size.
NOTE: SNPs sorted as in Table 1 according to the European ancestry combined meta-analysis P value per locus. Significant Ps are shown in bold (see text for criteria). Effect alleles were chosen to reflect an increased risk for low levels of HRV, hence β's are all negative.
*P value, allele, EAF, N from z-score weighted meta-analysis of all cohorts using METAL and β, s.e. from inverse-variance meta-analysis of only HF cohorts using GWAMA.
†β of participants of European ancestry differs significantly from that of participants from African-American (diff β = 0.044, P = 0.0012) or Hispanic/Latino ancestry (diff β = − 0.023, P = 0.0195).

and sympathetic blood pressure regulation through the baroreflex loops; (iii) mechanotransduction or intracellular pathways stimulated by sinoatrial stretch or (iv) the efficiency of the actual vagal gating process. These processes can have a strong impact on HRV, but less so on mean heart rate. We found six SNPs in four loci, including our top hit (rs12974991 in *NDUFA11*), that may act on the individual differences in these processes as they had no discernible effect on heart rate, in spite of their significant impact on HRV.

The genome-wide significant SNPs in *GNG11*, *RGS6* and *NEO1* were eQTLs and in strong LD with the top mQTLs and eQTLs for the corresponding genes. Two of these (*GNG11*, *RGS6*) readily provide a biological hypothesis to account for the associations detected in the meta-analysis. The C alleles of rs4262 and rs180238 of *GNG11* coding for the γ11 subunit of the heterotrimeric G-protein complex Gαβγ cause decreased expression of this subunit and were associated with lower HRV. The effects of the *GNG11* eQTLs associated with lower HRV are likely to lower the availability of the γ11 subunit, thereby reducing Gβγ component-induced GIRK activation. This potentially blunts the heart rate change in response to the oscillatory changes in cardiac vagal activity.

The regulator of heterotrimeric G-protein complex signalling, type 6 (*RGS6*) gene on chromosome 14 was found to be linked to three independent signals for SDNN and RMSSD. RGS6 acts as a critical negative regulator of M₂R signalling in the sinoatrial node of the heart rapidly terminating Gβγ signalling and thus curtailing vagal lowering of the heart rate[25,26]. The results of our meta-analysis are consistent with a role for *RGS6* in decreasing HRV previously hinted at by animal experimentation[23,27] and a human case report[27,28]. The T allele of our eQTL *RGS6* SNP

(rs4899412) causes increased expression of *RGS6*. By increasing *RGS6* expression, the T allele acts as a gain-of-function mutation that gives rise to a decrease in GIRK-channel signalling and the observed decrease in HRV. Of note, Rgs6$^{−/−}$ mice, that show the expected increase in HRV, are characterized by a strong bradycardia and an increased susceptibility to AV block and atrial fibrillation which is attributed to an enhancement of GIRK-induced sinoatrial and atrioventricular node hyperpolarization by removing the negative regulation of Gβγ by RGS6 (refs 23,26,28).

The association of the rs2680344 SNP in *HCN4* is puzzling because HCN signalling does not involve the fast M₂R-GIRK channels and cannot translate rapid vagal fluctuation into beat-to-beat variation in IBI length, that is, HRV. The effect of the *HCN4* SNP on HRV may be secondary to its effects on the average slope of the diastolic depolarization[29]. The HCN4 protein is a key component of the $I_f$ channel[30–32] that generates the pacemaker potential by a gradual depolarization of the sinoatrial myocyte cell membrane during diastole. This 'pacemaker depolarization' phase is known to be slowed by loss-of-function mutations in the *HCN4* that lead to lower heart rate[31] and the $I_f$ is the known site of action for ivabradine and other therapeutic agents used to slow heart rate in angina patients[32]. Of note, both ivabradine treatment[33] and loss-of-function mutations increase the risk for atrial fibrillation[34]. In contrast, gain-of-function mutations in the sensitivity of *HCN4* for cAMP lead to higher heart rate[30]. This leads us to hypothesize that the A allele of rs2680344 in *HCN4* either is itself a gain-of-function mutation or tags such a mutation because it increases heart rate[18].

High HRV is associated with lower morbidity and mortality in patients with cardiovascular disease[5], hypertension[7], end-stage

renal disease[8] and diabetes[9], but also in apparently healthy individuals[11,12]. Using LD score regression on meta-GWAS summary statistics from various risk factors and endpoints we find some evidence for overlap in the genetic variants causing low HRV and increased risk for disease, but significance was reached only for systolic and diastolic blood pressure after correction for multiple outcomes tested. These genetic correlations are compatible with causal effects of cardiac vagal control in the aetiology of disease, but they could also be ascribed to reversed causality, where the disease process leads to lower cardiac vagal control. A strength of this study in this regard is that analyses were confined to individuals in good cardiac health, that is, cohorts excluded patients with existing cardiovascular diseases or medication potentially impacting HRV. Because we selected individuals in good cardiac health reverse effects of disease on HRV seem less likely, although some latent pathology could have been present. However, an alternative explanation that is harder to rule out is that the genetic correlation derives from pleiotropic effects of genetic variants common to both outcomes.

Further strengths of this study were the consistency of results across the different HRV traits used to capture cardiac vagal control and the generalization of the HRV SNP effects to different ancestries, in spite of known ethnic differences in absolute resting HRV[35]. Results also held in men and women separately and across a very large range of mean cohort ages spanning from early childhood to the late middle ages; in spite of a strong reduction in HRV values with aging[36]. Although effects of age and sex on HRV were taken into account in the analyses, many other factors were not. The ideal design would have corrected for the known effects of respiration depth and rate on HRV, which are independent of vagal activity[37]. These could not be added as covariates because they were not available in most cohorts. We were liberal in excluding other covariates like BMI, smoking and exercise in the GWAS analyses. These traits are substantially heritable themselves and adjusting for heritable covariates can bias the genome-wide association effects[38] or even induce non-existing associations through collider bias[39]. Finally, instructions on pre-ECG recording behaviours like physical activity, and caffeine, alcohol or nicotine use were not rigorously standardized across cohorts.

Direct clinical relevance of most current GWAS findings is still low and our study is no exception. Potential future clinical use of our findings hinges on the ability of our genetic variants to capture (sub)cortical, brainstem and medullary transmission of tonic vagal activity to the sinoatrial node, not just the impact of that activity on heart rate. Subcortical generation of tonic vagal activity is an important biomarker for cardiovascular health and potentially modifiable by interventions on psychosocial stress[40] and lifestyle habits[41]. It can even be a transdiagnostic biomarker for psychopathology and executive cognitive functioning possibly by reflecting the integrity of prefrontal cortex functioning[42]. Genetic markers for HRV may prove useful as instrumental variables in Mendelian randomization[43] to test causal hypotheses on the effects of centrally generated vagal activity on behavioural and health outcomes.

In conclusion, this meta-analysis detects a critical role for genetic variation in Gβγ and HCN signalling in explaining individual differences in HRV. The HRV variants detected can help guide further investigations of the functional consequences and potential therapeutic implications of individual differences in sinoatrial Gβγ signalling.

## Methods

**Study cohorts.** Appropriate IRB approval and informed consent from participants in all participating cohorts was obtained. Full information on consent procedures and details of the IRB boards are provided in the Supplementary Note 8.

**HRV measurement.** In this study, we investigated three HRV traits: SDNN, the RMSSD and pvRSA or HF. SDNN and RMSSD were derived from the IBI time series obtained from the R waves in the ECG[4]. HF was calculated from Wavelet or Fourier decomposition with power obtained from a high frequency band of either 0.15–0.40 Hz or 0.15–0.50 Hz. A time domain measure of RSA was derived by pvRSA using a respiratory signal co-registered with the ECG. Estimates of pvRSA are obtained by subtracting the shortest IBI during heart rate acceleration in the inspiration phase from the longest IBI during heart rate deceleration in the expiration phase.

HRV traits were extracted from the IBI time series preferably based on 2–10 min periods of ECG in a standardized setting, at rest and in a sitting/supine position. If ambulatory data were available, we advised cohorts to extract a period of sitting still in the evening, when this proved feasible. Supplementary Table 2 lists the actual way HRV was assessed by the participating cohorts. For the cohorts analysed in stage 2, we extended our HRV measurements to include cohorts with 10 s and/or 20 s ECG recordings, as RMSSD and SDNN based on these ultra-short recordings have shown a good agreement with 4–5 min recordings[3]. Furthermore, since IBI time series require reliable detection of the R-wave only, a three-lead ECG was considered sufficient while the use of more leads was encouraged. For pvRSA, an additional respiration signal of sufficient quality to detect beginning and end of inspiration and expiration was needed.

SDNN and RMSSD have prevailed in epidemiological studies because they are more easily assessed in large cohorts and, as noted above, can be obtained even from short ECG recordings. HF and pvRSA were available in fewer cohorts, but they better reflect the cardiorespiratory coupling that drives the oscillatory modulation of vagal effects in the sinoatrial node. In the typical resting respiratory frequency range, these time- and frequency-domain measures of RSA are much less contaminated by oscillations in cardiac sympathetic control than SDNN (and other measures of HRV that span a broader frequency range). This is due to the temporal dynamics of the sinoatrial node signalling pathway that acts as a low pass filter allowing only oscillations in vagal effects to translate into HRV, whereas for sympathetic effects or vagal effects at progressively higher respiratory frequencies the node acts as a leaky integrator causing more tonic changes in heart rate[1]. Phasic modulation of vagal effects is therefore captured most purely by pvRSA or HF. Because pvRSA and HF are conceptually similar and highly correlated with each other ($r > 0.80$) across a wide range of values for respiration and heart rates[44], we grouped the analyses on pvRSA and HF under the label pvRSA/HF.

**Study population.** Cohorts that had data on at least one of the three HRV traits and genome-wide data were invited to participate in the first (discovery) stage of the Genetic Variance in Heart Rate Variability (V$_g$HRV) consortium. The stage 1 discovery analysis was performed in up to 28,700 individuals of European ancestry from a maximum of 20 cohorts. Independent cohorts with either genome-wide or gene-centred array data or with the ability to perform wet-lab genotyping on the single-nucleotide polymorphism (SNPs) taken forward from the first stage were included in the second (replication) stage. This stage included additional data from up to 24,474 individuals from 11 cohorts of European ancestry (see Supplementary Tables 1–4 for cohort descriptions and details).

**Association analysis: stage 1 (discovery).** The following exclusion criteria were applied a priori: (1) individuals with heart disease (for example, angina, past myocardial infarction, left ventricular failure) and (2) individuals known to use antidepressants (particularly tricyclic antidepressants) and all anticholinergic agents (for example, digoxin, atropine and acetylcholinesterase inhibitors) because of the strong effects that these drugs have on HRV. Individuals reporting over the counter use of anticholinergic agents were not excluded.

Imputation of SNPs was done to extend and create similar SNP databases between cohorts using different genotyping platforms. Most of the cohorts used the HapMap Phase II release 22 CEU panel as reference, but later releases (for example, release 24) or other reference data sets (for example, 1000 Genomes) were also used (Supplementary Table 4).

Each cohort performed linear regression analyses on all available HRV traits using an additive SNP model adjusting for age at the time of ECG recording, sex, principal components—to adjust for population stratification—and other study-specific parameters; all HRV traits were log-transformed because of the skewness of their distributions. Only autosomal associations were examined. Analyses were performed for all individuals as well as for men and women separately.

**Stage 1 meta-analysis.** Before meta-analysis, quality control of all uploaded cohort files was performed using the QCGWAS package[45]. In case of issues the cohorts were notified and problems were solved. Using the QCGWAS results, specific imputation quality and allele frequency thresholds were set for each cohort.

An inverse-variance, fixed-effects meta-analysis was performed for RMSSD and SDNN for which SNPs of the different cohorts were merged based on rs-id. For pvRSA/HF, we performed a sample size weighted meta-analysis using z-scores with METAL[46], since we combined results of two HRV phenotypes (pvRSA and HF) that have different units and ranges, and therefore incomparable SNP effect sizes. To get an idea of the size of the SNP effect on pvRSA/HF, we obtained effect sizes

and s.e.'s from an additional fixed-effect meta-analysis on the GWAS results of the (majority of) cohorts that measured HF. Results of the meta-analyses were double genomic control corrected[47] to control for potential inflation as a result of population stratification within and between cohorts. The results included all SNPs that met the following selection criteria: (a) a minor allele frequency in the meta-analysis of >1%, and (b) present in at least one third of the cohorts. This resulted in 2,555,913 SNPs being analysed for SDNN, 2,534,714 SNPs for RMSSD and 2,628,894 SNPs for pvRSA/HF. For each trait separately, SNPs with a $P < 1 \times 10^{-6}$ were clumped for LD using pairwise LD checking in SNAP[48] to ascertain independent primary and secondary signals ($r^2 < 0.1$). A total of 23 lead SNPs in 14 loci were selected for follow-up in the second (replication) stage.

**Stage 2 meta-analysis.** Stage 2 cohorts applied the same exclusion criteria and performed the same association analysis as in the discovery stage, but analyses were restricted to the 23 lead SNPs. If a SNP was not available in a cohort, the best available proxy was used instead based on strongest LD according to the 1000 Genomes database. To verify homogeneity of the results in the stage 2 cohorts with those in the stage 1 cohorts, the stage 1 meta-analysis effect sizes of the 23 SNPs were correlated to the effect sizes obtained in each cohort for each of the HRV traits. If a negative correlation ($r < 0$) was found, the cohort/trait pair was excluded from stage 2 analysis. For this reason results from one cohort for SDNN were excluded. The replication results were then meta-analysed per trait using an inverse-variance fixed-effects meta-analysis for RMSSD and SDNN and a sample size $P$ weighted meta-analysis using $z$-scores in METAL[46] for pvRSA/HF. SNPs were matched based on rs-id. Next the association results from both stages were combined in the same way. A SNP was only considered to be significantly associated to HRV if it satisfied the following criteria: (1) it had $P < 1 \times 10^{-6}$ in stage 1, (2) it had a one-sided $P < 0.05$ in the stage 2 meta-analysis congruent with the direction of effect in the stage 1 meta-analysis and (3) it had a genome-wide significant $P < 5 \times 10^{-8}/3$ (two-sided) in the combined meta-analysis of stage 1 and 2 results, correcting for the testing of three separate traits.

**Conditional analysis.** In the discovery stage independent SNPs were selected for follow-up based on LD clumping ($r^2 < 0.1$). To confirm independence between these SNPs within the loci on chromosome 14 and 15, we applied the conditional-and-joint analysis as implemented in the Genome-wide Complex Trait Analysis software package[49] to the stage 1 summary statistics of RMSSD and SDNN with the genotype data of the NESDA cohort[50] of 1,925 individuals as the LD reference data set. In addition, cohort-level individual data on log-transformed RMSSD and SDNN of 12,101 individuals from the Dutch Lifelines cohort[51] were analysed using linear regression analysis with age and sex as covariates conditioned on the other associated SNP(s) within the locus.

**Gene-based association analysis (VEGAS).** We performed gene-based testing with the full set of $\sim 2.5$ M HapMap SNPs from GWAS results of all three phenotypes, using VEGAS (Supplementary Table 17). This software has the advantage of accounting for LD structure and the possibility to define a range beyond the gene bounds to include promoter, 5′UTR, intronic and 3′UTR regions into the analysis. We defined a 50 kb extra window beyond the genes, considered every SNP in this window for the gene-based analysis, and ran the analyses per chromosome with up to $10^6$ permutations. A $P < 2.5 \times 10^{-6}$ ($= 0.05/\sim 20,000$ genes) was considered as the threshold for significance.

**Variance explained.** The Lifelines and NESDA cohorts were used for genetic risk score and polygenic risk score analyses to determine the percentage of variance explained by independent HRV SNPs that were genome-wide significant, and by SNPs meeting increasingly lenient significance thresholds, respectively. Lifelines and NESDA represent examples of a population-based cohort and a cohort ascertained on case-control status (for major depressive disorder). Both recruited adult participants. To test the stability of explained variance across the life span, we repeated this analysis in two other Dutch cohorts, the adolescent TRAILS-Pop cohort[52] (age 10–18) and the ABCD cohort consisting of young children (age 5–7) (ref. 53).

For the genetic risk score, stage $1+2$ summary statistics were used for the selection of HRV SNPs. No correction was needed for ABCD as genotyping in this cohort had finished only after completion of the meta-analyses. However, the NESDA cohort had been included in both stage 1 and 2, TRAILS-Pop in stage 1, and Lifelines in stage 2, so the effect sizes and s.e.'s of the HRV SNPs were corrected to subtract the effects of those cohorts to obtain independent validation cohorts[54]. Also, only SNPs were used in the genetic risk score if they remained genome-wide significant after analytically subtracting these cohort's effects from the meta-analysis. Genetic risk scores of the remaining SNPs (Lifelines: SDNN(9), RMSSD(7), pvRSA/HF(5); NESDA: SDNN(9), RMSSD(11), pvRSA/HF(5); TRAILS-Pop: SDNN(10), RMSSD(11), pvRSA/HF(5)) weighted by the adjusted effect size were calculated for the participants of all four cohorts and regressed on the three HRV traits (pvRSA/HF was not available in Lifelines). Explained variance was computed as the change in $R^2$ from a model with and without the genetic risk score, while adjusting both for age, sex and principal components.

To compute the polygenic risk scores, the imputed genotypes were first converted to best-guess genotypes. This was done regardless of the imputation quality, since it was previously shown that even low-quality SNPs might contribute to the variance explained by SNPs (ref. 54). The SNP set was further pruned for LD using PriorityPruner (http://prioritypruner.sourceforge.net/) to select independent SNPs, taking the significance of the SNP in the discovery meta-analysis of each of the HRV traits into account. This provided three LD-pruned SNP sets. Polygenic risk scores were then calculated in PLINK[55] using significance thresholds of $5 \times 10^{-8}$, $5 \times 10^{-7}$, $5 \times 10^{-6}$, $5 \times 10^{-5}$, $5 \times 10^{-4}$, 0.005, 0.05, 0.5 and 1 and associated with the three HRV traits and resting heart rate in the Lifelines, NESDA, TRAILS-Pop and ABCD cohorts. For NESDA and TRAILS-Pop pruning and polygenic risk score analysis was based on analytically corrected results, since these cohorts were part of stage 1 of our study[54].

**Heritabilities and genetic correlations.** We applied genomic restricted maximum likelihood analysis implemented in the Genomic Complex Trait Analysis software package[56] in the Lifelines cohort (Supplementary Table 18) to estimate the percentages of additive phenotypic variance that can be explained by common SNPs (that is, common SNP heritability denoted as $h^2_{SNP}$). For this analysis, SNPs from the HapMap Phase 3 project were selected to obtain a set of independent SNPs. We further used LD score regression to estimate the heritabilities of the three HRV traits and the genetic correlation among HRV traits and with heart rate[20]. The GWAS meta-analysis summary statistics for RMSSD, SDNN and pvRSA/HF were obtained from stage 1 of the current study, and the GWAS meta-analysis summary statistics for heart rate from the discovery stage of a recent GWAS meta-analysis for heart rate[18]. The LD scores required by the method were computed using 1000 Genomes data of Europeans. The heritabilities of the three HRV measurements were estimated using the univariate model of this method. Cross-phenotype LD score regression analysis was performed using the LDSC tool (LD SCore) to estimate genetic correlations between pairs of phenotypes[20].

In addition, we used the Oman Family Study[57] to perform univariate and bivariate analyses in five multigenerational highly inbred pedigrees to estimate the heritabilities for and the genetic correlations between log-transformed RMSSD, SDNN, HF and heart rate using SOLAR (v7.2.5) (ref. 58).

**Generalization to other ethnicities.** We further examined the generalization of loci identified after meta-analysis of stage 1 and 2 results to other ethnicities using data from 11,234 individuals of two Hispanic/Latino cohorts, and 6,899 individuals from five African-American cohorts (Supplementary Tables 1–4). Stage 3 meta-analyses were performed in the same way as in stage 2 of this study to assess the effect of the HRV-associated SNPs in individuals of Hispanic/Latino and African-American ancestry, in the combined set of European and Hispanic/Latino ancestry, in the combined set of European and African-American ancestry, and in all three ethnicities combined. Here, we applied the same criteria for significance as in stage 2 described above, that is, a SNP was only considered to be significantly associated to HRV if: (1) it had $P < 1 \times 10^{-6}$ in stage 1 meta-analysis in European individuals, (2) it had a one-sided $P < 0.05$ in the new ethnicity specific meta-analysis congruent with the direction of effect in the stage 1 meta-analysis in European individuals and (3) it had a genome-wide significant $P < 5 \times 10^{-8}/3$ (two-sided) in the combined meta-analysis.

**Correcting HRV for heart rate.** The well-known inverse association between HRV and heart rate in part reflects a dependency of the variance in IBI on the mean IBI that is unrelated to cardiac vagal activity[59]. That is, the slower the heart rate, the longer the IBI, and therefore, any proportionally minor beat-to-beat differences in IBI are more pronounced at slower heart rates. This occurs on top of the well-established dual effect of cardiac vagal activity that lowers heart rate and increases HRV[15,16]. Although these two mechanisms (biological, mean-variance dependency) are impossible to completely separate, we conducted three analyses to test whether the HRV SNPs were robust to correction for the mean IBI.

First, we corrected SDNN and RMSSD for their dependency on mean IBI by using the coefficient of variation, which is a more parsimonious solution[19] than the logarithmic approach suggested by Monfredi et al.[29]. We obtained the summary statistics for the resting heart rate GWAS meta-analysis[18] from: https://walker05.u.hpc.mssm.edu/ and used the GWIS procedure[17] to infer a GWA analysis of the coefficient of variation of the SDNN and the RMSSD. We approximated the coefficients of variation by (SDNN/X) × 100% and (RMSSD/X) × 100%, respectively, where X equals 60,000 per heart rate. Transformation from heart rate to IBI is required as both terms in the coefficient of variation (HRV and IBI) are in milliseconds, whereas the heart rate GWAS meta-analysis used heart rate in beats per minute. As the coefficients of variation were skewed, we used a log-transformation. As an example of the linear approximation by GWIS we assume that the increaser effect of one allele for an SDNN SNP is $+0.2$ with the same SNP reducing heart rate by $-0.1$. Given a mean SDNN of 100 and mean heart rate of 60, we can then approximate (omitting some nuances adequately explained in Nieuwboer et al.[17]) the effect of the SNP on

the coefficient of variation of the SDNN as:

$$\ln\left(\frac{100 + 0.2}{60,000/(60 - 0.1)}\right) - \ln\left(\frac{100}{60,000/(60)}\right) = 0.00033$$

We used the delta method to approximate a s.e. for the effect of the SNP given that we know the s.d.'s for the SNP effects on SDNN and HR, and their dependence. We obtain the dependence from analysis with LD score regression[20].

Second, we performed association analyses for our 17 top SNPs on the actual log-transformed coefficients of variation of SDNN and RMSSD computed in the Lifelines, NESDA and TRAILS-Pop cohorts and then meta-analysed these results. Because pvRSA and HF are expressed on different scales, such a meta-analysis was not feasible for pvRSA/HF.

Third, we repeated the association analysis for our 17 top SNPs on SDNN, RMSSD and pvRSA/HF in the Lifelines, NESDA and TRAILS-Pop cohorts with and without adjusting for heart rate as a covariate and performed mediation tests with the Sobel test to assess the mediation effect of heart rate on the HRV SNP association. Significance of the Sobel $t$-value was determined using a bootstrap procedure ($n = 10,000$ permutations). The mediation $P$ values of the three cohorts for SDNN and RMSSD and two for pvRSA/HF (as this was not available in Lifelines) were next meta-analysed to determine the significance of mediation and to compute the percentage of the SNP effect on HRV that was mediated through its effects on heart rate. We note that this is likely an overcorrection because the HRV SNPs are expected to influence heart rate through a common biological mechanism, that is, changes in cardiac vagal activity.

**Association of the HRV SNPs with heart rate.** We conducted a lookup of the 17 (11 independent) HRV lead SNPs identified in this study using the results of a recent GWAS meta-analysis for heart rate[18]. A HRV-associated SNP was considered to be significantly associated with resting heart rate if the GWAS meta-analysis result for heart rate was $< 0.05/11 = 0.0045$. Three separate HRV weighted multi-SNP genetic risk scores were calculated from 10 (SDNN), 11 (RMSSD) and five (pvRSA/HF) HRV SNPs, respectively, (based on all genome-wide significant SNPs for the respective HRV trait in the stage $1 + 2$ meta-analysis). These were tested for their effect $\alpha$ on resting heart rate using the gtx package in R (https://cran.r-project.org/web/packages/gtx), which approximated $\alpha$ by $(\Sigma\omega \times \beta \times se_{\beta}^{-2})/(\Sigma\omega^2 \times se_{\beta}^{-2})$ with $se_{\alpha} \cong \sqrt{(1/\Sigma\omega^2 \times se_{\beta}^{-2})}$, where $\omega$ is the effect of the SNP on HRV, $\beta$ is the effect of the SNP on heart rate and $se_{\beta}$ is the s.e. of $\beta$. This approximation requires only single SNP association summary statistics extracted from GWAS results[60]. The effects of the multi-SNP genetic risk scores were considered as statistically significant when the $P$ was less than 0.0045 (correcting for 11 traits; heart rate and the 10 cardiometabolic traits described below).

In addition to these lookups that were restricted to genome-wide significant SNPs, we employed LD score regression[20] that uses the full summary statistics of the HRV and heart rate GWAS meta-analyses to compute genetic correlations.

We further examined the variance in resting heart rate explained by multi-SNP genetic risk scores (based on the lead SNPs only) and of the full polygenic risk scores for HRV in the four Dutch cohorts Lifelines, NESDA, TRAILS-Pop and ABCD. The identical approach was used as done previously for the computation of variance explained in the HRV traits themselves.

**Association of heart rate SNPs with HRV.** We also performed reverse analyses to detect the effects of heart rate SNPs on the HRV traits. In our GWAS meta-analysis results for SDNN, RMSSD and pvRSA/HF, we performed a lookup for the 21 previously identified heart rate SNPs by den Hoed et al.[18]. A heart rate associated SNP was considered to be significantly associated with HRV if the $P$ was $< 0.05/21 = 0.0024$. The 21 heart rate SNPs were tested in a multi-SNP risk score for their effect on the HRV traits using the gtx approach as described above.

To examine the variance explained in the HRV traits by the 21 heart rate SNPs, multi-SNP genetic risk scores and polygenic risk scores based on the heart rate SNPs were computed in the Lifelines, NESDA, TRAILS-Pop and ABCD cohorts and these were tested for association with the available HRV traits. For the multi-SNP genetic risk scores weights were either the original SNP effect sizes on heart rate (for NESDA, TRAILS-Pop and ABCD) or corrected because of participation of the cohort in the GWAS meta-analysis (Lifelines). Only 15 of the 21 SNPs were used in the Lifelines cohort because five SNPs lost genome-wide significance after subtracting the SNP effects of the Lifelines cohort. One other SNP (rs826838) was removed because it was in LD ($r^2 = 0.15$) in Lifelines with a more significant heart rate SNP (rs7980799).

**Association with cardiometabolic traits and diseases.** We estimated the joint effect of the HRV SNPs on cardiometabolic and renal disease traits and endpoints. The traits included were systolic and diastolic blood pressure, body mass index and urinary albumin excretion as well as estimated glomerular filtration rate based on creatinine. The clinical outcomes used were heart failure, coronary artery disease, atrial fibrillation, sudden cardiac death and type 2 diabetes. The relevant consortia (Supplementary Table 11) and/or corresponding authors of the studies were contacted with the request to perform a lookup and provide summary GWAS meta-analysis results for our list of 17 SNPs.

The association analyses consisted of the same three steps as used for heart rate. First, we checked the $P$ of our HRV SNPs (or their proxies) in the cardiometabolic trait or disease GWAS meta-analysis results. Second, three separate HRV weighted genetic risk scores were calculated from 11 (RMSSD), 10 (SDNN) and five (pvRSA/HF) HRV SNPs, respectively (based on all genome-wide significant SNPs for the respective HRV trait in the stage $1 + 2$ meta-analysis). These were tested for their effect on the clinical outcomes using a regression model in the gtx package in R as described above for the association of the HRV SNPs with heart rate. The effects of the genetic risk scores were considered as statistically significant when the $P$ was less than 0.0045 (0.05/11, correcting for heart rate and the 10 traits and diseases).

In addition to these lookups that were restricted to genome-wide significant SNPs, we employed LD score regression[20] that uses the full GWAS summary statistics of the HRV and cardiometabolic traits and diseases to compute genetic correlations.

**Search for known functional SNPs (in silico annotation).** We followed an in silico bioinformatics-based approach[61] to search and annotate SNPs in the regions surrounding the 17 identified HRV SNPs. For this purpose SNP positions were converted from National Center for Biotechnology Information (NCBI) build 36, Human Genome 18, to NCBI build 37, Human Genome 19, (GRCh37/hg19) using the NCBI Genome Remapping service tool (http://www.ncbi.nlm.nih.gov/genome/tools/remap). For $\pm 1$ Mb regions surrounding the SNPs, we downloaded the according variance call format file from the 1000 Genomes Project. We used data of 503 European ancestry individuals from 1000 Genomes Project Phase 3 (version 5.a.) to calculate LD between the HRV SNP and all other SNPs within the area. SNPs in moderate to high LD ($r^2 \geq 0.5$) were subsequently selected and annotated by ANNOVAR software[62] for functionality. For all non-synonymous SNPs loss-of-function and gain-of-function was determined by using the SIFT and PolyPhen prediction scores. A SNP was categorized as deleterious if the SIFT score was $\leq 0.05$ or the PolyPhen score was between 0.957 and 1 (probably damaging).

We used RegulomeDB to integrate results from the RoadMap Epigenomics and ENCODE projects to identify variants that are likely to have functional consequences using the lead SNPs identified for the three HRV traits. We distilled information on transcription factor binding and chromatin states for SNPs that showed most evidence of being functional, that is, for SNPs with a RegulomeDB score $< 4$.

Finally, all the HRV SNPs and those that were in high LD ($r^2 \geq 0.8$) with them were looked up in the National Human Genome Research Institute GWAS catalogue to check for association with other complex traits or diseases identified in previous GWAS studies[63].

**eQTL analyses.** We performed expression quantitative trait locus (eQTL) analysis in whole blood to identify regulatory variants that were associated with the HRV SNPs using the gene-expression database from NESDA[50] and NTR[64] cohorts. The sample used for this analysis consisted of 4,896 individuals of European ancestry. For complete details on the sample and the procedures, see[65].

eQTL effects were tested with a linear model approach using MatrixeQTL[66] with expression level as dependent variable and SNP genotype values as independent variable. In this study we only tested cis effects for our HRV SNPs, meaning that the probe was at a distance $< 1$ Mb from the SNP on the genome according to GRCh37/hg19. For each probe set that displayed a statistically significant association with at least one SNP in the cis region, we identified the most significantly associated SNP (top eQTL). Conditional eQTL analysis was carried out by first residualizing probe set expression using the corresponding top eQTL and then repeating the eQTL analysis using the residualized data.

All HRV SNPs with significant results in the NESDA/NTR eQTL data were looked up in two other independent whole blood eQTL databases, eQTLs in lymphoblastoid cell lines, eQTLs in 10 different brain regions, and a heart eQTL database.

**mQTL analyses.** We obtained mQTL results from a previously published study[67]. In short, genome-wide DNA methylation data were generated using Illumina 450 k arrays for 3,841 whole blood samples. Corresponding genotype data were imputed using the Genome of The Netherlands[68] reference panel. To determine the effect of nearby genetic variation on methylation levels (cis-mQTLs) a Spearman rank correlation and corresponding $P$ value was computed for each CpG-SNP pair, in which the CpG and SNP location were no further than 250 kb apart. To control for multiple testing, we used a permutation procedure to empirically control the false discovery rate at 5%. The distribution of observed $P$ values was compared to the $P$ value distribution obtained from the analyses on permuted data. For a permutation the sample identifiers of the genotype data set were shuffled, breaking the link between the genotype data set and the methylation data set. This was repeated 10 times to obtain a stable distribution of $P$ values under the null hypothesis. To determine the false discovery rate only the strongest effect per CpG in both the real analysis and in the permutations were selected.

**Gene prioritization using four bioinformatics approaches.** Potentially causal genes for the associations identified by GWAS were identified using four previously

described bioinformatics tools: ToppGene, Endeavour, MetaRanker and DEPICT (Supplementary Table 19). To this end, we first retrieved positional coordinates for all lead SNPs according to GRCh37/hg19. These coordinates were used to extract all genes located within ± 40 kb of lead SNPs using the UCSC genome browser. The identified genes subsequently served as input for ToppGene and Endeavour, together with two genes with established roles in sinus node function (*HCN4*) and synaptic signal transmission (*ACHE*) that served as training genes. For MetaRanker, we first combined results of the stage 1 + 2 meta-analyses of GWAS for the three HRV traits, retained the association with the lowest *P* for lead SNPs that were identified for multiple traits, and subsequently provided SNPs, *P* values, and the same two test genes (*HCN4* and *ACHE*) as input. For DEPICT—arguably the most powerful and informative of the four methods—we used results from the stage 1 meta-analysis for all SNPs that reached a *P* for association $<10^{-5}$ as input, for each of the three HRV outcomes separately. In order for genes to be prioritized by the combined four approaches, they needed to be either: (1) selected by DEPICT for at least one of the three HRV outcomes; or (2) identified by at least two of the three remaining tools (ToppGene, Endeavour and/or MetaRanker).

**Network and functional enrichment analyses.** We performed gene network and enrichment analysis using the GeneMANIA algorithm, which uses data resources on genetic interactions, protein–protein, co-expression, shared protein domains and co-localization networks. To build a functional interaction network, we selected genes as input for this analysis using the following criteria: (a) genes implicated by gene prioritization using the four bioinformatics approaches described above, (b) the genes closest to our 17 HRV SNPs, (c) genes to which linked ($r^2 > 0.50$) non-synonymous SNPs mapped, (d) genes to which other linked ($r^2 > 0.80$) SNPs mapped, (e) genes identified by VEGAS, and (f) expression probe gene names significantly associated with HRV eQTLs (false discovery rate $<0.01$). The input gene list was extended to 100 by their most strongly interacting genes and a weighted composite functional association network was constructed[61]. Subsequently, functional enrichment analysis of all genes of the constructed interaction network against Gene Ontology (GO) terms was performed to find the most enriched GO terms (Supplementary Table 20). Significantly enriched GO terms (false discovery rate $<0.10$) were visualized as highlighted boxes within their corresponding GO tree depicted by the RamiGO R package[69] (Supplementary Fig. 10).

**Tissue and gene-set enrichment analyses.** We used DEPICT for a tissue enrichment analysis to tabulate tissues that are enriched for expression of genes located within ± 40 kb of SNPs with a $P < 10^{-5}$ association with the HRV traits. DEPICT calculates the likelihood of every known gene to be a member of, amongst others, KEGG, GEO or REACTOME-based gene sets ($N = 14,461$) to create reconstituted gene sets. It then determines which of these reconstituted gene sets are enriched for the HRV genes. A graphical representation of DEPICT's reconstituted gene-set enrichment analysis ($P < 0.05$ after Bonferroni correction for examining three HRV gene sets) was generated using a script that is based on an affinity propagation clustering algorithm by Frey *et al.*[70]. Interactions between gene sets are considered significant if the Pearson coefficient, which is based on the number of genes that are shared between gene sets, is $>0.3$.

**Data availability.** Summary statistics of the meta-analyses are available on request from the corresponding authors after a formal data access request procedure and approval by the VgHRV consortium.

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

## Acknowledgements

A full list of acknowledgements appears in the Supplementary Information. Funding sources had no involvement in the collection, analysis and interpretation of the data.

## Author contributions

H.S. and E.d.G. designed and overviewed the project; I.M.N., M.L.M. and V.T. did the quality control of the individual GWA results and performed the meta-analyses; C.Al., D.G.B., P.I.W.d.B., R.B., D.B., P.T.E., O.H.F., M.A.G., C.A., A.H., H.H., M.J., M.Ku., C.C.L., C.M.L., A.P.M., G.N., D.T.O., J.O., A.P., B.M.P., O.T.R., V.B.R., J.A.S., K.S., J.-C.T., A.T., A.B.Z., D.C., M.K.E., P.v.d.H., M.H., E.I., M.-R.J., S.K., M.Kä., C.K., D.K., T.L., L.L., C.M.N., C.J.O., A.J.O., B.P., A.P.R., H.R., J.D.R., J.I.R., T.S., B.H.S., H.T., T.G.M.V., F.W.A., B.J.J.M.B., E.A.W., M.d.H., H.S. and E.d.G. were Principal Investigators of the participating cohorts; M.L.M., R.J., D.J., A.K., A.Mi., J.F.T., S.A., C.Al., A.Al., J.A., R.J.B., D.B., A.R.B., A.D., M.E., J.S.F., P.F., M.G.J.G., M.A.G., V.G., H.H., N.H.-K., X.J., J.J., M.J., A.M.K., J.A.K., T.K., J.D.L., D.L., M.C.L., M.Ne., K.N., D.T.O., S.P., A.P., B.M.P., O.T.R., V.B.R., M.S., D.S., J.H.S., N.L.S., E.Z.S., N.S., J.A.S., P.K.S., K.S.-S., J.S., C.A.S., A.Vo., G.W., Z.-M.Z., H.K.G., M.H., E.I., S.K., M.Kä., M.Ki., D.K., T.L., L.L., C.M.N., H.R., A.M.v.R., T.G.M.V., B.J.J.M.B., S.R.H., E.A.W. and E.d.G. performed HRV phenotyping; R.J., P.G., L.-P.L., A.X.M., M.M.-N., E.S., A.W., M.H.Z., A.Ab., T.A., M.B., M.G.J.G., J.-J.H., X.J., J.J., M.Ku., Y.Li., C.M.L., H.M.z.S., Y.M., N.M., A.P.M., M.A.N., D.T.O., D.S., J.H.S., A.S., K.D.T., L.E.T., A.G.U., M.W., K.C.W., A.B.Z., M.K.E., M.-R.J, M.Kä., D.K., T.L., C.M.N., C.J.O., H.R., J.D.R., T.G.M.V. and H.S. performed genotyping and imputation; I.M.N., M.L.M., V.T., A.T.A., R.J., A.Va, B.v.d.H., C.L.A., J.C.B., B.D., J.v.D., S.M.G., P.G., J.H, V.H., S.-J.H., D.J., K.F.K., A.K, B.P.K., J.K., S.W.v.d.L., L.-P.L., A.X.M., A.Mi., P.J.v.d.M., M.M.-N., M.Ni., E.S., J.D.S., J.F.T., N.V., A.W., D.Z., M.H.Z., A.Ab., F.A., J.A., P.I.W.d.B., M.B., G.B., A.R.B., I.C., G.B.E., J.F.F., J.S.F., D.H., J-J.H, A.M.K., T.K., J.D.L., Y.Li, H.J.L., C.M.L., S.A.L., A.Ma., B.M., Y.M., A.P.M., M.A.N., K.E.N., V.B.R., J.A.S., P.K.S., A.M.S., K.S.-S, T.A.T., J.v.S., A.Vo., Q.W., C.M.N. and A.M.v.R. performed data analysis; I.M.N., M.L.M., V.T., S.R.H., E.A.W., M.d.H., H.S. and E.d.G. drafted and edited the manuscript. All authors contributed to and critically reviewed the manuscript.

## Additional information

**Competing interests:** M.A.G. has an equity interest in San Diego Instruments. B.M.P. serves on the DSMB of a clinical trial funded by the manufacturer (Zoll LifeCor) and on the Steering Committee of the Yale Open Data Access Project funded by Johnson & Johnson. The remaining authors declare no competing financial interests.

Ilja M. Nolte[1,*], M. Loretto Munoz[1,*], Vinicius Tragante[2,*], Azmeraw T. Amare[1,3,4], Rick Jansen[5], Ahmad Vaez[1,6], Benedikt von der Heyde[7,8], Christy L. Avery[9], Joshua C. Bis[10], Bram Dierckx[11,12], Jenny van Dongen[13], Stephanie M. Gogarten[14], Philippe Goyette[15], Jussi Hernesniemi[16,17,18], Ville Huikari[19], Shih-Jen Hwang[20,21], Deepali Jaju[22], Kathleen F. Kerr[14],

Alexander Kluttig[23], Bouwe P. Krijthe[24], Jitender Kumar[7,8], Sander W. van der Laan[25], Leo-Pekka Lyytikäinen[16,17], Adam X. Maihofer[26,27], Arpi Minassian[26,27], Peter J. van der Most[1], Martina Müller-Nurasyid[28,29,30], Michel Nivard[13,31], Erika Salvi[32], James D. Stewart[9,33], Julian F. Thayer[34], Niek Verweij[35], Andrew Wong[36], Delilah Zabaneh[37,38], Mohammad H. Zafarmand[39,40], Abdel Abdellaoui[13,31], Sulayma Albarwani[41], Christine Albert[42], Alvaro Alonso[43], Foram Ashar[44], Juha Auvinen[19,45], Tomas Axelsson[46], Dewleen G. Baker[27,26], Paul I.W. de Bakker[47,48], Matteo Barcella[32], Riad Bayoumi[49], Rob J. Bieringa[1], Dorret Boomsma[13,31], Gabrielle Boucher[15], Annie R. Britton[50], Ingrid E. Christophersen[51,52,53], Andrea Dietrich[54], George B. Ehret[55,56], Patrick T. Ellinor[52,57], Markku Eskola[18,58], Janine F. Felix[24], John S. Floras[59,60], Oscar H. Franco[24], Peter Friberg[61], Maaike G.J. Gademan[39], Mark A. Geyer[26], Vilmantas Giedraitis[62], Catharina A. Hartman[63], Daiane Hemerich[2,64], Albert Hofman[24], Jouke-Jan Hottenga[13,31], Heikki Huikuri[65], Nina Hutri-Kähönen[66,67], Xavier Jouven[68], Juhani Junttila[65], Markus Juonala[69,70], Antti M. Kiviniemi[65], Jan A. Kors[71], Meena Kumari[50,72], Tatiana Kuznetsova[73], Cathy C. Laurie[14], Joop D. Lefrandt[74], Yong Li[75], Yun Li[76,77,78], Duanping Liao[79], Marian C. Limacher[80], Henry J. Lin[81,82], Cecilia M. Lindgren[83,84], Steven A. Lubitz[52,57], Anubha Mahajan[84], Barbara McKnight[10,14,85], Henriette Meyer zu Schwabedissen[86], Yuri Milaneschi[5], Nina Mononen[16,17], Andrew P. Morris[84,87], Mike A. Nalls[88], Gerjan Navis[89], Melanie Neijts[13,31], Kjell Nikus[18,90], Kari E. North[9,91], Daniel T. O'Connor[92], Johan Ormel[63], Siegfried Perz[93], Annette Peters[30,93,94], Bruce M. Psaty[10,95,96], Olli T. Raitakari[97,98], Victoria B. Risbrough[26,27], Moritz F. Sinner[29,30], David Siscovick[99], Johannes H. Smit[5], Nicholas L. Smith[96,100,101], Elsayed Z. Soliman[102], Nona Sotoodehnia[103], Jan A. Staessen[73], Phyllis K. Stein[104], Adrienne M. Stilp[14], Katarzyna Stolarz-Skrzypek[105], Konstantin Strauch[28,106], Johan Sundström[107], Cees A. Swenne[108], Ann-Christine Syvänen[46], Jean-Claude Tardif[15,109], Kent D. Taylor[110], Alexander Teumer[111], Timothy A. Thornton[14], Lesley E. Tinker[85], André G. Uitterlinden[24,112,113], Jessica van Setten[2], Andreas Voss[114], Melanie Waldenberger[93,115], Kirk C. Wilhelmsen[116,117], Gonneke Willemsen[13,31], Quenna Wong[14], Zhu-Ming Zhang[102,118], Alan B. Zonderman[119], Daniele Cusi[120,121], Michele K. Evans[119], Halina K. Greiser[122], Pim van der Harst[35], Mohammad Hassan[41], Erik Ingelsson[7,8,123], Marjo-Riitta Järvelin[19,45,124,125], Stefan Kääb[29,30], Mika Kähönen[126,127], Mika Kivimaki[50], Charles Kooperberg[85], Diana Kuh[36], Terho Lehtimäki[16,17], Lars Lind[107], Caroline M. Nievergelt[26,27], Chris J. O'Donnell[20,21,128], Albertine J. Oldehinkel[63], Brenda Penninx[5], Alexander P. Reiner[85,100], Harriëtte Riese[63], Arie M. van Roon[74], John D. Rioux[15,109], Jerome I. Rotter[110], Tamar Sofer[14], Bruno H. Stricker[24,129], Henning Tiemeier[11,24], Tanja G.M. Vrijkotte[39], Folkert W. Asselbergs[2,130,131], Bianca J.J.M Brundel[132], Susan R. Heckbert[10,100], Eric A. Whitsel[9,133], Marcel den Hoed[7,8], Harold Snieder[1],* & Eco J.C. de Geus[13,31],*

[1]Department of Epidemiology, University of Groningen, University Medical Center Groningen, PO Box 30001, Groningen 9700 RB, The Netherlands. [2]Department of Cardiology, Division Heart and Lungs, University Medical Center Utrecht, Heidelberglaan 100, Utrecht 3584CX, The Netherlands. [3]Department of Epidemiology, School of Medicine, University of Adelaide, Adelaide, South Australia 5005, Australia. [4]College of Medicine and Health Sciences, Bahir Dar University, Bahir Dar 6000, Ethiopia. [5]Department of Psychiatry, EMGO Institute for Health and Care Research and Neuroscience Campus Amsterdam, VU University Medical Center/GGZ inGeest, Amsterdam 1081 BT, The Netherlands. [6]School of Medicine, Isfahan University of Medical Sciences, Isfahan 81746-73461, Iran. [7]Department of Medical Sciences, Molecular Epidemiology, Uppsala University, Uppsala 75237, Sweden. [8]Science for Life Laboratory, Uppsala University, Uppsala 75237, Sweden. [9]Department of Epidemiology, Gillings School of Global Public Health, University of North Carolina, Chapel Hill, North Carolina 27599, USA. [10]Cardiovascular Health Research Unit, Department of Medicine, University of Washington, Seattle, Washington 98104, USA. [11]Department of Child and Adolescent Psychiatry/Psychology, Department of Child and Adolescent Psychiatry, PO Box 2060, Rotterdam 3000 CB, The Netherlands. [12]The Generation R Study Group, Erasmus MC, PO Box 2060, Rotterdam 3000 CB, The Netherlands. [13]Department of Biological Psychology, Behavioral and Movement Sciences, VU University, Amsterdam 1081 BT, The Netherlands. [14]Department of Biostatistics, School of Public Health, University of Washington, Seattle, Washington 98195, USA. [15]Montreal Heart Institute, Montreal, Quebec, Canada H1T 1C8. [16]Department of Clinical Chemistry, Fimlab Laboratories, Tampere 33520, Finland. [17]Department of Clinical Chemistry, University of Tampere School of Medicine, Tampere 33014, Finland. [18]Department of Cardiology, Heart Hospital, Tampere University Hospital, Tampere 33521, Finland. [19]Center for Life Course Health Research, University of Oulu, Oulu 90014, Finland. [20]Framingham Heart Study, Framingham, Massachusetts 01702, USA. [21]Population Sciences Branch, National Heart, Lung and Blood Institute, Bethesda, Maryland 20892, USA. [22]Department of Clinical Physiology, Sultan Qaboos University Hospital, Muscat—Al Khoudh 123, Sultanate of Oman. [23]Institute of Medical Epidemiology, Biostatistics and Informatics, Martin-Luther-University Halle-Wittenberg, Halle (Saale) 06097, Germany. [24]Department of Epidemiology, Erasmus MC, University Medical Center Rotterdam, PO Box 2060, Rotterdam 3000 CB, The Netherlands. [25]Laboratory of Experimental Cardiology, Department of Heart and Lung, University Medical Center Utrecht, Heidelberglaan 100, Utrecht 3584 CX, The Netherlands. [26]Department of Psychiatry, University of California, San Diego, San Diego, California 92093, USA. [27]Center for Stress and Mental Health (CESAMH), VA San Diego Healthcare System, San Diego, California 92161, USA. [28]Institute of Genetic Epidemiology, Helmholtz Zentrum München—German Research Center for Environmental Health, Neuherberg 85764, Germany. [29]Department of Medicine, University Hospital Munich, Ludwig-Maximilians-University, Munich 80539, Germany. [30]DZHK (German Centre for Cardiovascular Research), Partner site Munich Heart Alliance, Munich 80336, Germany. [31]EMGO + Institute for Health and Care Research, VU University & VU University Medical Center, Amsterdam 1081 HV, The Netherlands. [32]Department of Health Sciences, University of Milano, Milano 20122, Italy. [33]Carolina Population Center, University of North Carolina, Chapel Hill, North Carolina 27599, USA. [34]Department of Psychology, The Ohio State University, 1835 Neil Avenue, Columbus, Ohio 43210, USA. [35]Department of Cardiology, University of Groningen, University Medical Center Groningen, PO Box 30001, Groningen 9700 RB, The Netherlands. [36]MRC Unit for Lifelong Health and Ageing, University College London, 33 Bedford Place, London WC1B 5JU, UK. [37]Institute of Psychiatry, Psychology & Neuroscience, King's College London,

De Crespigny Park, London SE5 8AF, UK. [38] University College London Genetics Institute, University College London, London WC1E 6BT, UK. [39] Department of Public Health, Academic Medical Center (AMC), University of Amsterdam, Amsterdam 1105 AZ, The Netherlands. [40] Department of Obstetrics and Gynaecology, Academic Medical Centre, University of Amsterdam, Amsterdam 1105 AZ, The Netherlands. [41] Department of Physiology, College of Medicine and Health Sciences, Sultan Qaboos University, Muscat Al-Khoudh 123, Sultanate of Oman. [42] Brigham and Women's Hospital, Harvard Medical School, Boston, Massachusetts 02115, USA. [43] Department of Epidemiology, Rollins School of Public Health, Emory University, Atlanta, Georgia 30322, USA. [44] McKusick-Nathans Institute of Genetic Medicine, Johns Hopkins University School of Medicine, Baltimore, Maryland 21205, USA. [45] Unit of Primary Health Care, Oulu University Hospital, Oulu 90220, Finland. [46] Department of Medical Sciences, Molecular Medicine, Uppsala University, Uppsala 75237, Sweden. [47] Department of Genetics, Center for Molecular Medicine, University Medical Center Utrecht, Utrecht 3584 CX, The Netherlands. [48] Department of Epidemiology, Julius Center for Health Sciences and Primary Care, University Medical Center Utrecht, Utrecht 3584 CX, The Netherlands. [49] College of Medicine, Mohammed Bin Rashid University, PO Box 505055, Dubai Healthcare City 66566, United Arab Emirates. [50] Department of Epidemiology and Public Health, University College London, London WC1E 6BT, UK. [51] Cardiovascular Research Center, Massachusetts General Hospital, Boston, Massachusetts 02114, USA. [52] Program in Medical and Population Genetics, The Broad Institute of Harvard and MIT, Cambridge, Massachusetts 02114, USA. [53] Department of Medical Research, Bærum Hospital, Vestre Viken Hospital Trust, Rud 1346, Norway. [54] Department of Child and Adolescent Psychiatry, University of Groningen, University Medical Center Groningen, PO Box 30001, Groningen 9700 RB, The Netherlands. [55] Center for Complex Disease Genomics, McKusick-Nathans Institute of Genetic Medicine, Johns Hopkins University School of Medicine, Baltimore, Maryland 21205, USA. [56] Cardiology, Department of Specialties of Internal Medicine, Geneva University Hospital, Geneva 1211, Switzerland. [57] Cardiac Arrhythmia Service & Cardiovascular Research Center, Massachusetts General Hospital, Boston, Massachusetts 02114, USA. [58] Department of Cardiology, University of Tampere School of Medicine, Tampere 33014, Finland. [59] University Health Network and Mount Sinai Hospital Division of Cardiology, Department of Medicine, University of Toronto, Ontario, Canada M5S. [60] Toronto General Research Institute, University Health Network, Toronto, Ontario, Canada M5G 2C4. [61] Department of Molecular and Clinical Medicine, Institute of Medicine, Sahlgrenska University Hospital, University of Gothenburg, Gothenburg SE-413 45, Sweden. [62] Department of Public Health and Caring Sciences, Molecular Geriatrics, Uppsala University, Uppsala 75237, Sweden. [63] Interdisciplinary Center Psychopathology and Emotion regulation, Department of Psychiatry, University of Groningen, University Medical Center Groningen, PO Box 30001, Groningen 9700 RB, The Netherlands. [64] CAPES Foundation, Ministry of Education of Brazil, Brasília DF 70040-020, Brazil. [65] Research Unit of Internal Medicine, Medical Research Center Oulu, Oulu University Hospital and University of Oulu, Oulu 90220, Finland. [66] Department of Pediatrics, Tampere University Hospital, Tampere 33521, Finland. [67] Department of Pediatrics, University of Tampere School of Medicine, Tampere 33014, Finland. [68] INSERM U970, Paris Descartes University, Paris 75006, France. [69] Department of Medicine, University of Turku, Turku 20520, Finland. [70] Division of Medicine, Turku University Hospital, Turku 20521, Finland. [71] Department of Medical Informatics, Erasmus Medical Center, Rotterdam 3015 CE, The Netherlands. [72] ISER, Essex University, Colchester, Essex CO4 3SQ, UK. [73] Studies Coordinating Centre, Research Unit Hypertension and Cardiovascular Epidemiology, KU Leuven Department of Cardiovascular Sciences, University of Leuven, Leuven 3000, Belgium. [74] Department of Internal Medicine, Division of Vascular Medicine, University of Groningen, University Medical Center Groningen, PO Box 30001, Groningen 9700 RB, The Netherlands. [75] Division of Genetic Epidemiology, Institute for Medical Biometry and Statistics, Medical Center—University of Freiburg, Faculty of Medicine, University of Freiburg, Freiburg 79110, Germany. [76] Department of Genetics, University of North Carolina, Chapel Hill, North Carolina 27599, USA. [77] Department of Biostatistics, University of North Carolina, Chapel Hill, North Carolina 27599, USA. [78] Department of Computer Science, University of North Carolina, Chapel Hill, North Carolina 27599, USA. [79] Division of Epidemiology, Department of Public Health Sciences, Penn State University College of Medicine, Hershey, Pennsylvania 17033, USA. [80] Division of Cardiovascular Medicine, University of Florida College of Medicine, Gainesville, Florida 32611, USA. [81] Institute for Translational Genomics and Population Sciences, Department of Pediatrics, Los Angeles Biomedical Research Institute at Harbor-UCLA Medical Center, Torrance, California 90502, USA. [82] Division of Medical Genetics, Department of Pediatrics, Harbor-UCLA Medical Center, Torrance, California 90502, USA. [83] Li Ka Shing Centre for Health Information and Discovery, The Big Data Institute, University of Oxford, Oxford OX3 7BN, UK. [84] Wellcome Trust Centre for Human Genetics, University of Oxford, Oxford OX3 7BN, UK. [85] Public Health Sciences Division, Fred Hutchinson Cancer Research Center, Seattle, Washington 98109, USA. [86] Biopharmacy, Department Pharmaceutical Sciences, University of Basel, Basel CH-4056, Switzerland. [87] Department of Biostatistics, University of Liverpool, Liverpool L69 3GL, UK. [88] Laboratory of Neurogenetics, National Institute on Aging, National Institutes of Health, Bethesda, Maryland 20892, USA. [89] Department of Internal Medicine, Division of Nephrology, University of Groningen, University Medical Center Groningen, PO Box 30001, Groningen 9700 RB, The Netherlands. [90] Department of Cardiology, University of Tampere, School of Medicine, Tampere 33014, Finland. [91] Carolina Center for Genome Sciences, University of North Carolina, Chapel Hill, North Carolina 27599, USA. [92] Department of Medicine, University of California, San Diego, San Diego, California 92093, USA. [93] Institute of Epidemiology II, Helmholtz Zentrum München—German Research Center for Environmental Health, Neuherberg 85764, Germany. [94] German Center for Diabetes Research, Neuherberg 85764, Germany. [95] Departments of Epidemiology and Health Services, University of Washington, Seattle, Washington 98195, USA. [96] Group Health Research Institute, Group Health Cooperative, Seattle, Washington 98101, USA. [97] Department of Clinical Physiology and Nuclear Medicine, Turku University Hospital, Turku 20521, Finland. [98] Research Centre of Applied and Preventive Cardiovascular Medicine, University of Turku, Turku 20520, Finland. [99] The New York Academy of Medicine, New York, New York 10029, USA. [100] Department of Epidemiology, School of Public Health, University of Washington, Seattle, Washington 98195, USA. [101] Seattle Epidemiologic Research and Information Center, Veterans Affairs Office of Research and Development, Seattle, Washington 98108, USA. [102] Epidemiological Cardiology Research Center (EPICARE), Division of Public Health Sciences, Wake Forest School of Medicine, Winston-Salem, North Carolina 27157, USA. [103] Cardiovascular Health Research Unit, Division of Cardiology, Departments of Medicine and Epidemiology, University of Washington, Seattle, Washington 98101, USA. [104] Heart Rate Variability Lab, Washington University School of Medicine, St Louis, Missouri 63108, USA. [105] First Department of Cardiology, Interventional Electrocardiology and Hypertension, Jagiellonian University Medical College, Cracow 31-008, Poland. [106] Institute of Medical Informatics, Biometry and Epidemiology, Chair of Genetic Epidemiology, Ludwig-Maximilians-Universität, Munich 81377, Germany. [107] Department of Medical Sciences, Cardiovascular Epidemiology, Uppsala University, Uppsala 751 85, Sweden. [108] Department of Cardiology, Leiden University Medical Center, Leiden 2300 RC, The Netherlands. [109] Université de Montréal, Montreal, Quebec, Canada H3T IJ4. [110] Institute for Translational Genomics and Population Sciences, Departments of Pediatrics and Medicine, Los Angeles Biomedical Research Institute at Harbor-UCLA Medical Center, Torrance, California 90509, USA. [111] Institute for Community Medicine, University Medicine Greifswald, Greifswald 17475, Germany. [112] Department of Internal Medicine, Erasmus University Medical Center, Rotterdam 3015 CE, The Netherlands. [113] Netherlands Genomics Initiative (NGI)-sponsored Netherlands Consortium for Healthy Aging NCHA), Leiden 2300 RC, The Netherlands. [114] Institute of Innovative Health Technologies—IGHT Jena Ernst-Abbe-Hochschule Jena, Jena 07745, Germany. [115] Research Unit of Molecular Epidemiology, Helmholtz Zentrum München—German Research Center for Environmental Health, Neuherberg 85764, Germany. [116] Departments of Genetics and Neurology University of North Carolina, Chapel Hill, North Carolina 27599, USA. [117] The Renaissance Computing Institute, Chapel Hill, North Carolina 27599, USA. [118] Department of Epidemiology & Prevention, Division of Public Health Sciences, Wake Forest School of Medicine, Winston-Salem, North Carolina 27157, USA. [119] Laboratory of Epidemiology and Population Sciences, National Institute on Aging, National Institutes of Health, Baltimore, Maryland 21224, USA. [120] Institute of Biomedical Technologies, CNR—Italian National Research Council, Milan 20090, Italy. [121] KOS Genetic SRL, Bresso (Milano) 20091, Italy. [122] German Cancer Research Centre, Division of Cancer Epidemiology, Heidelberg 69210, Germany. [123] Department of Medicine, Division of Cardiovascular Medicine, Stanford University School of Medicine, Stanford, California 94305, USA. [124] Department of Epidemiology and Biostatistics, School of Public Health, Faculty of Medicine, St Mary's campus, Imperial

College London, London W2 1PG, UK. [125] Biocenter Oulu University of Oulu, Oulu 90014, Finland. [126] Department of Clinical Physiology, Tampere University Hospital, Tampere 33521, Finland. [127] Department of Clinical Physiology, University of Tampere, School of Medicine, Tampere 33014, Finland. [128] Cardiology Section, Boston Veteran's Administration Healthcare, Boston, Maryland 02132, USA. [129] Inspectorate for Health Care, The Hague 2511 VX, The Netherlands. [130] Institute of Cardiovascular Science, University College London, 222 Euston Road, London NW1 2DA, UK. [131] Durrer Center for Cardiogenetic Research, ICIN-Netherlands Heart Institute, Utrecht 3501 DG, The Netherlands. [132] Department of Physiology, Institute for Cardiovascular Research, VU University Medical Center, De Boelelaan 1118, Amsterdam 1081 HV, The Netherlands. [133] Department of Medicine, University of North Carolina, Chapel Hill, North Carolina 27599, USA. * These authors contributed equally to this work.

DOI: 10.1038/ncomms16140 OPEN

# Erratum: Genetic loci associated with heart rate variability and their effects on cardiac disease risk

Ilja M. Nolte et al.[#]

#A full list of authors and their affiliations appears at the end of the paper.

*Nature Communications* 8:15805 doi: 10.1038/ncomms15805 (2017); Published 14 Jun 2017; Updated 2 Aug 2017

In Supplementary Fig. 10 of this Article, images for panels a and b were inadvertently omitted. The correct version of Supplementary Fig. 10 is provided as Supplementary Information associated with this Erratum. Furthermore, the affiliation details for Azmeraw T. Amare, Benedikt von der Heyde, and Marcel den Hoed are incorrect in this Article. The correct affiliation details for these authors are given below.

Azmeraw T. Amare:

Department of Epidemiology, University of Groningen, University Medical Center Groningen, PO Box 30001, Groningen 9700 RB, The Netherlands.

School of Medicine, University of Adelaide, Adelaide, South Australia, SA 5005, Australia

College of Medicine and Health Sciences, Bahir Dar University, Bahir Dar 6000, Ethiopia.

Benedikt von der Heyde:

Department of Immunology, Genetics and Pathology, Medical Genetics and Genomics, Uppsala University, Uppsala 75327, Sweden.

Science for Life Laboratory, Uppsala University, Uppsala 75237, Sweden.

Marcel den Hoed:

Department of Immunology, Genetics and Pathology, Medical Genetics and Genomics, Uppsala University, Uppsala 75327, Sweden.

Science for Life Laboratory, Uppsala University, Uppsala 75237, Sweden.

Ilja M. Nolte, M. Loretto Munoz, Vinicius Tragante, Azmeraw T. Amare, Rick Jansen, Ahmad Vaez, Benedikt von der Heyde, Christy L. Avery, Joshua C. Bis, Bram Dierckx, Jenny van Dongen, Stephanie M. Gogarten, Philippe Goyette, Jussi Hernesniemi, Ville Huikari, Shih-Jen Hwang, Deepali Jaju, Kathleen F. Kerr, Alexander Kluttig, Bouwe P. Krijthe, Jitender Kumar, Sander W. van der Laan, Leo-Pekka Lyytikäinen, Adam X. Maihofer, Arpi Minassian, Peter J. van der Most, Martina Müller-Nurasyid, Michel Nivard, Erika Salvi, James D. Stewart, Julian F. Thayer, Niek Verweij, Andrew Wong, Delilah Zabaneh, Mohammad H. Zafarmand, Abdel Abdellaoui, Sulayma Albarwani, Christine Albert, Alvaro Alonso, Foram Ashar, Juha Auvinen, Tomas Axelsson, Dewleen G. Baker, Paul I.W. de Bakker, Matteo Barcella, Riad Bayoumi, Rob J. Bieringa, Dorret Boomsma, Gabrielle Boucher, Annie R. Britton, Ingrid Christophersen, Andrea Dietrich, George B. Ehret, Patrick T. Ellinor,

Markku Eskola, Janine F. Felix, John S. Floras, Oscar H. Franco, Peter Friberg, Maaike G.J. Gademan, Mark A. Geyer, Vilmantas Giedraitis, Catharina A. Hartman, Daiane Hemerich, Albert Hofman, Jouke-Jan Hottenga, Heikki Huikuri, Nina Hutri-Kähönen, Xavier Jouven, Juhani Junttila, Markus Juonala, Antti M. Kiviniemi, Jan A. Kors, Meena Kumari, Tatiana Kuznetsova, Cathy C. Laurie, Joop D. Lefrandt, Yong Li, Yun Li, Duanping Liao, Marian C. Limacher, Henry J. Lin, Cecilia M. Lindgren, Steven A. Lubitz, Anubha Mahajan, Barbara McKnight, Henriette Meyer zu Schwabedissen, Yuri Milaneschi, Nina Mononen, Andrew P. Morris, Mike A. Nalls, Gerjan Navis, Melanie Neijts, Kjell Nikus, Kari E. North, Daniel T. O'Connor, Johan Ormel, Siegfried Perz, Annette Peters, Bruce M. Psaty, Olli T. Raitakari, Victoria B. Risbrough, Moritz F. Sinner, David Siscovick, Johannes H. Smit, Nicholas L. Smith, Elsayed Z. Soliman, Nona Sotoodehnia, Jan A. Staessen, Phyllis K. Stein, Adrienne M. Stilp, Katarzyna Stolarz-Skrzypek, Konstantin Strauch, Johan Sundström, Cees A. Swenne, Ann-Christine Syvänen, Jean-Claude Tardif, Kent D. Taylor, Alexander Teumer, Timothy A. Thornton, Lesley E. Tinker, André G. Uitterlinden, Jessica van Setten, Andreas Voss, Melanie Waldenberger, Kirk C. Wilhelmsen, Gonneke Willemsen, Quenna Wong, Zhu-Ming Zhang, Alan B. Zonderman, Daniele Cusi, Michele K. Evans, Halina K. Greiser, Pim van der Harst, Mohammad Hassan, Erik Ingelsson, Marjo-Riitta Järvelin, Stefan Kääb, Mika Kähönen, Mika Kivimaki, Charles Kooperberg, Diana Kuh, Terho Lehtimäki, Lars Lind, Caroline M. Nievergelt, Chris J. O'Donnell, Albertine J. Oldehinkel, Brenda Penninx, Alexander P. Reiner, Harriëtte Riese, Arie M. van Roon, John D. Rioux, Jerome I. Rotter, Tamar Sofer, Bruno H. Stricker, Henning Tiemeier, Tanja G.M. Vrijkotte, Folkert W. Asselbergs, Bianca J.J.M Brundel, Susan R. Heckbert, Eric A. Whitsel, Marcel den Hoed, Harold Snieder & Eco J.C. de Geus

