## [Peer Review File · Nature Communications]

Reviewer #1 (Remarks to the Author):

The authors have done a comprehensive study of collecting and analyzing heart rate variability (HRV) data and further downstream bioinformatic analyses. They identified 17 independent SNPs in 8 loci. It comes as no surprise that the heart rate variability loci overlap considerably with heart rate loci. The multi-SNP risk score analysis showed that lower HRV predicted lower risk of atrial fibrillation (AF) and type 2 diabetes (T2D).

Some questions that I would like answered in the discussion perhaps:

1. The opening comment on how reduced vagal control (low HRV) is associated with increased risk for cardiac morbidity & mortality. I am not sure if I could get any sense if it is true towards the end of the paper. For most cardiometabolic traits (where public data is available), the risk score analysis is negative. The 2 significant findings were: multi-snp risk score for HRV SNPs (lower RMSSD & SDNN) show lower risk of atrial fibrillation (AF) and lower HRV was associated with lower risk of T2D. Later on, the authors mention a higher HRV in *Rgs6*^{-/-} mice increased susceptibility to AV block & AF. It is a bit confusing.
2. The variability in heart rate is when the control of sympathetic and/or para-sympathetic nervous system gets disturbed. Both these autonomic fibres are tonically active and there are many factors which affect this fine tuning and thereby variation in heart rate. There is mention that the cohorts in this study were healthy and did not take any medication affecting heart rate. How about caffeine intake before the time of measurements? This will increase the basal heart rate and dampen the variation in heart rate. Do you have any data to confirm ?
3. The variation in SDNN is pretty vast (average ranges from ~20 to 760?) TRAILS-CC shows the maximum. The forest plot does not show the study as an outlier so probably it is not alarming?
4. How relevant is blood as a cell type to make any conclusions on functional gene?
5. I personally find forest plots easier to follow if they are sorted by effect size.
6. I know that UK Biobank are measuring ECG and numerous other heart metrics but am not aware whether the data on heart rate variability is available yet or not? It will certainly add a lot of power to your results (with the unfortunate cost of delaying this paper).

Reviewer #2 (Remarks to the Author):

The presented study contains a meta-analysis of genome-wide association studies for heart rate variability. In a two-stage procedure, the authors identify eight chromosomal loci associated with at least one of the three investigated heart rate variability traits, i.e., standard deviation of the normal-to-normal inter beat intervals, root mean square of the successive differences of inter beat intervals, and peak-valley respiratory sinus arrhythmia or high frequency power. As proteins encoded by genes that have been identified in this analysis are involved in cardiac pacemaker function, the authors claim to provide mechanistic insight into the functional biology underlying the genetic association.

The study is the result of a large effort. The authors are well qualified for this type of analysis and the analysis plan is linearly conceived. There are, however, a few issues that need to be addressed.

Major comments

1. This study is the thus far largest meta-analysis on this topic. To this reviewer, no comparable study is known. The primary claims, i.e., the association of the reported chromosomal loci with heart rate variability measures, are convincing. In this light, further analyses aiming to elucidate a functional role of the detected variants are speculative and of uncertain value (e.g., blood eQTLs rather than cardiomyocyte eQTLs). Thus, the “Potential functional HRV variants” and “Network and functional enrichment analyses” sections as well as the corresponding Discussion sections should be shortened.
2. The authors mention the association of heart rate variability with a number of diseases. Here, a heart rate variability loci multi-SNP risk score was only associated with atrial fibrillation but not coronary artery disease, type 2 diabetes... In this reviewer’s opinion, this indicates that heart rate variability changes are indeed rather a consequence of the particular diseases. This should be briefly discussed.
3. The most interesting finding in this study to this reviewer is the association of variants in the GNG11, RGS6, and HCN4 genes with heart rate variability. The paper would be strengthened by functional data directly investigating a mechanistic role of the identified SNPs as discussed by the authors (p12, l273-p13, l281).
4. The Discussion is very much focused on the potential pathophysiological influence of the variants. The Discussion would benefit from a focus on rather the genetic findings than speculative implications.

Minor comments

1. Clear written English. Many abbreviations making the paper difficult to read. Careful revision suggested.
2. Clear tables. Figure 1 ok. Figure 2 confusing, speculative, should be revised.

Reviewer #3 (Remarks to the Author):

This manuscript details a large scale, two stage GWAS of three heart rate variability (HRV) traits followed by some in silico functional annotation. HRV is associated with increased risk of sudden cardiac death and all-cause mortality, and has previously been shown to be heritable, suggesting an important genetic component but a large scale GWAS of these traits has not been previously conducted. The authors perform a large number of statistical analyses and find associations in 8 loci, several of which have been previously associated with heart rate and are biologically plausible candidates for regulating HRV, but additional ones that may be independent of HR.

The strengths of the study are in the very large sample size (up to N=53000) with a two-staged discovery and validation and subsequent meta-analysis approach, the somewhat novel phenotype that has been incompletely evaluated at a genetic level, evaluation in non-Caucasian populations, triangulating results to assess prediction of GWAS identified loci with incident events using genotype risk scores, and some in silico functional analysis (with the caveats that that information was not particularly informative).

There are however, several issues that temper enthusiasm for the study. The results are likely only incremental in our understanding of cardiac sinus node regulation as many of the loci have already been shown to be associated with heart rate itself and thus the results likely represent the fact that these variants regulate heart rate and it is known that resting heart rate is correlated with HRV. Thus, the authors should adjust for resting heart rate in all of their analyses to provide a context for true novel findings for HRV vs. those known to affect sinus node regulation through their known effects on resting HR. The authors did do a "look up" in GWAS of heart rate to pull p-values for their HRV loci, but this is insufficient to address this issue. The other major concern is the heterogeneity of the ways HRV was assessed in each cohort including cohorts that included very short tracings. At the minimum, sensitivity analyses of cohorts that have a full 24 hours of tracings to determine HRV should be done. More minor issues are the lack of functional data, particularly for the novel loci and the large number of analyses that make the manuscript somewhat difficult to follow (adding a figure of study flow with what adjustments for multiple comparisons were made at each stage would be helpful to address this).

Some more minor questions/issues:

- 1) The authors need to provide justification for their p-value threshold which is not corrected for the number of traits analyzed.
- 2) How correlated were the three HRV traits in their data?
- 3) What were the imputation scores for the top loci?
- 4) Only a small amount of variance in the trait is explained by their loci; although not unexpected the authors should comment on the clinical significance of that.
- 5) Please include OR for prediction of risk for incident events.
- 6) The first paragraph on page 10 is difficult to understand and it is not clear what set of analyses those represent. Was the GSEA done using the RNA microarray data from the subset that had that done on whole blood for HRV traits themselves?
- 7) Given the linear models used, did the HRV traits meet criteria for parametric analyses after log transformation? This is particularly necessary for the SOLAR analyses that are very sensitive to deviations in normality.
- 8) Was the genomic inflation rate for each stage?
- 9) Please include a reference for the “double genomic control correction”
- 10)

Rebecca Furlong
Senior Editor
Nature Communications

Dear Dr. Furlong,

Thank you for the invitation to revise and resubmit our manuscript “Identification of genetic loci associated with heart rate variability and their effects on heart rate and cardiac disease risk.” to Nature Communications. We were very pleased with the overall positive reception of this *first ever* GWA meta-analysis (N=53,174) on heart rate variability (HRV), a trait widely used in cardiology to characterize vagal control of heart rate for which no robustly associated loci had previously been identified.

The reviewers’ comments have provided excellent guidance and we believe this revised version is a clear improvement over the original. We are grateful to the reviewers for their constructive comments.

Three major changes were made that can be summarized as:

- As suggested by the reviewers, we reduced the sections on functional *in silico* annotations of the HRV variants as well as the corresponding Discussion sections. These were indeed inherently somewhat speculative.
- We added a number of additional analyses to address reviewers’ concerns, including the full genetic correlations between HRV and the disease risk factors/endpoints [by going back to the various consortia and retrieving their full GWAS summary stats], a series of analyses addressing the dependency of HRV on HR, and a sensitivity analysis testing potential laboratory/ambulatory heterogeneity in the HRV phenotypes across participating cohorts.
- We have improved our discussion on the potential functional relevance of the genetic variants and illustrated in Figure 2 by better indicating which elements were based on existing knowledge on sinoatrial node signalling and which on the new hypothesized effects of the three genetic variants that were identified in the current study.

Many additional changes were made in this revision. A full point-by-point reply to the reviewers and your editorial request is given in a separate document. Resulting changes in the manuscript have all been highlighted.

We again thank you for handling our manuscript and your willingness to consider it for publication after satisfactory revision.

Yours sincerely,

Ilja Nolte, Harold Snieder and Eco de Geus
on behalf of the VgHRV consortium

EDITORIAL REQUEST

Please ensure that the following requirements are met, and any relevant checklist is completed and uploaded as a Related Manuscript file type with the revised article.

REPLY

The revision closely follows the reporting requirements in:

http://www.nature.com/article-assets/npg/ncomms/authors/ncomms_lifesciences_checklist.pdf

and the format requirements in:

http://www.nature.com/article-assets/npg/ncomms/authors/ncomms_manuscript_checklist.pdf

REVIEWERS REMARKS TO THE AUTHORS

Reviewer #1

The authors have done a comprehensive study of collecting and analysing heart rate variability (HRV) data and further downstream bioinformatics analyses. They identified 17 independent SNPs in 8 loci. It comes as no surprise that the heart rate variability loci overlap considerably with heart rate loci. The multi-SNP risk score analysis showed that lower HRV predicted lower risk of atrial fibrillation (AF) and type 2 diabetes (T2D).

Some questions that I would like answered in the discussion perhaps:

COMMENT

1. The opening comment on how reduced vagal control (low HRV) is associated with increased risk for cardiac morbidity & mortality. I am not sure if I could get any sense if it is true towards the end of the paper. For most cardiometabolic traits (where public data is available), the risk score analysis is negative.

REPLY

You are correct; this was confusing. Absence of an association likely arose because we only requested the consortia to look up effects on the disease phenotypes for our 17 genome-wide SNPs for HRV. Although this risk score analysis makes good sense, the variance explained in HRV by these top SNPs is small, and HRV will only be one of many risk factors for these disease outcomes. This approach therefore has modest power to detect an association between the HRV SNPs and the disease outcomes, and failure to find the association is not prove of its absence. For the revision we re-contacted the original consortia for these disease risk factors and endpoints with a request to provide us with the full GWAS summary statistics so that we could compute the genetic correlation between HRV and these disease risk factors/endpoints using LD Score regression. We have added the genetic correlations to the results section with full details in Supplementary Table 11. As you will see the genetic correlations systematically pointed to an overlap in the genetic variants causing low HRV and increased risk for disease (i.e., negative correlations with e.g. SBP, DBP, Coronary Artery Disease, Heart Failure, Sudden Cardiac Death, BMI, and T2D) although significance was reached only for blood pressure after correction for number of outcomes tested. Atrial fibrillation was still positively correlated to low HRV, but we find the correlation unconvincing. Also the heterogeneity tests in *gtx* are now considered suboptimal, and we removed these tests. LD Score regression results are more telling than just the plots of the non-heterogeneous top SNPs in previous Supplementary Figures 7 and 8. We removed these figures and had to retrace our steps

regarding a clear AF – HRV connection, which we now consider too weak to interpret. We adapted the discussion to reflect this developing wisdom.

“Using LD Score regression on meta-GWAS summary statistics from various risk factors and endpoints we find some evidence for overlap in the genetic variants causing low HRV and increased risk for disease, but significance was reached only for systolic and diastolic blood pressure after correction for multiple outcomes tested. These genetic correlations are compatible with causal effects of cardiac vagal control in the etiology of disease, but they could also be ascribed to reversed causality, where the disease process leads to lower cardiac vagal control. A strength of this study in this regard is that analyses were confined to individuals in good cardiac health, i.e. cohorts excluded patients with existing cardiovascular diseases or medication potentially impacting HRV. Because we selected individuals in good cardiac health reverse effects of disease on HRV seems less likely, although some latent pathology could have been present. However, an alternative explanation that is harder to rule out is that the genetic correlation derives from pleiotropic effects of genetic variants common to both outcomes.”

COMMENT

The 2 significant findings were: multi-snp risk score for HRV SNPs (lower RMSSD & SDNN) show lower risk of atrial fibrillation (AF) and lower HRV was associated with lower risk of T2D. Later on, the authors mention a higher HRV in Rgs6^{-/-} mice increased susceptibility to AV block & AF. It is a bit confusing.

REPLY

Based on the added results on the disease risk factors/endpoints using LD Score regression we can no longer maintain that there is a clear AF – HRV connection. The results of the genetic correlation between HRV and T2D, although only marginally significant for SDNN ($r=-0.26$, $p=0.01$), are now much more in line with the expectation based on the studies showing a phenotype of lower HRV in diabetes patients.

COMMENT

2. The variability in heart rate is when the control of sympathetic and/or para-sympathetic nervous system gets disturbed. Both these autonomic fibres are tonically active and there are many factors which affect this fine tuning and thereby variation in heart rate. There is mention that the cohorts in this study were healthy and did not take any medication affecting heart rate. How about caffeine intake before the time of measurements? This will increase the basal heart rate and dampen the variation in heart rate. Do you have any data to confirm?

REPLY

Systematic information on pre-experimental use of either caffeine or nicotine was not gathered by all cohorts. We acknowledge potential effects of these substances on HRV but also note that the basal idea of GWAS meta-analysis is to be robust against differences in exact phenotyping. Furthermore, caffeine seems to influence heart rate mainly through sympathetic (and not vagal) effects, illustrated amongst others by direct sympathetic nerve recordings after caffeine infusion¹. Nonetheless, to address the concern of the reviewer we added the lack of strict pre-recording standardization on caffeine use, alcohol, smoking, and physical activity as a limitation of this study. *“Finally, instructions on pre-ECG recording behaviors like physical activity, and caffeine, alcohol, or nicotine use were not rigorously standardized across cohorts.”*

COMMENT

3. The variation in SDNN is pretty vast (average ranges from ~20 to 760?) TRAILS-CC shows the maximum. The forest plot does not show the study as an outlier so probably it is not alarming?

REPLY

This is an astute observation by the reviewer and we apologize for our own oversight. It led us to revisit the TRAILS-CC cohort. An unfortunate mistake was made in the prepared data by this cohort where we misplaced the decimal point for SDNN, which means it was a factor 10 too high. The cohort means have now been corrected in the Supplementary Table 3.

Note that this had no effect on the GWA results because we used log SDNN in this analysis. The factor 10 disappears in the intercept:

$\log(10 \cdot \text{SDNN}) = \log(10) + \log(\text{SDNN})$, leaving estimates of betas as reported in Tables and Forest plots correct.

COMMENT

4. How relevant is blood as a cell type to make any conclusions on functional gene?

REPLY

This is of course a valid point, and only future research will provide the answer in full. We certainly do not contest that large eQTL databases on heart tissue, or ideally SA and AV nodes would certainly have been preferred, but such databases are currently small (for heart, see our Koopmans and GTEx analysis in Supplementary Table 15) or non-existent (for SA node). That does not, however, mean that the identified eQTLs in blood cannot reflect the presence of an eQTL in another more relevant tissue. For many disease traits, including psychiatric disorders^{2, 3}, the SNPs identified with GWAS do contain eQTLs in blood even if the primary organ affected is clearly not hematological. In keeping, it has been shown that eQTLs are only partly tissue specific: almost 55% of blood eQTLs are also heart eQTLs for example⁴ (see Figure 2B from this reference below).

Figure 2B from⁴. From the legend: 'Dendrogram and heat map of pairwise eQTL sharing ... $\pi_1 = \text{Pr}(\text{eQTL in tissue } i \text{ given an eQTL in tissue } j)$ '

Our whole blood eQTL examinations in three different independent databases were well powered as these make up the largest eQTL databases in humans. Please note in Supplementary Table 15 that we detect eQTLs for *RGS6*, *HCN* and *GNG11* in the discovery database (NESDA/NTR) and then replicate all three in either a second (BIOS) or third (Westra et al.) blood eQTL database. We also replicate the *GNG11* blood eQTL in two other tissues (tibial artery and medulla).

COMMENT

5. I personally find forest plots easier to follow if they are sorted by effect size.

REPLY

We did consider various ways of sorting and believe the one used here is optimal. A clear disadvantage of the proposed ordering by the reviewer is that the cohorts would be ranked differently for each SNP. This makes it, e.g. much harder to detect whether a single cohort or cohorts with a specific choice of HRV phenotyping (for instance using 24-hours) somehow stand out from all other cohorts across more than one of the lead SNPs.

COMMENT

6. I know that UK Biobank are measuring ECG and numerous other heart metrics but am not aware whether the data on heart rate variability is available yet or not? It will certainly add a lot of power to your results (with the unfortunate cost of delaying this paper).

REPLY

Indeed, UKB has not done this and *de novo* scoring of the raw ECG data files to obtain the HRV metrics used in our own cohorts is non-trivial. Even so, our more important concern is that the UKB measured the ECG only during a bicycle exercise test, and its recovery phase, which are inherently poorly standardized. Unfortunately, no clearly defined sitting/supine resting ECG recordings are available. We could extract ~15 secs of stable ECGs before biking onset in some subjects suggesting that they were seated on the bike before actual pedaling started. Such a baseline is known to contain a strong anticipatory (fight-flight) activation of the autonomic nervous system and can be expected to deviate rather strongly from our resting ECG phenotype.

Reviewer #2

COMMENT

The presented study contains a meta-analysis of genome-wide association studies for heart rate variability. In a two-stage procedure, the authors identify eight chromosomal loci associated with at least one of the three investigated heart rate variability traits, i.e., standard deviation of the normal-to-normal inter beat intervals, root mean square of the successive differences of inter beat intervals, and peak-valley respiratory sinus arrhythmia or high frequency power. As proteins encoded by genes that have been identified in this analysis are involved in cardiac pacemaker function, the authors claim to provide mechanistic insight into the functional biology underlying the genetic association.

The study is the result of a large effort. The authors are well qualified for this type of analysis and the analysis plan is linearly conceived. There are, however, a few issues that need to be addressed.

COMMENT

1. This study is the thus far largest meta-analysis on this topic. To this reviewer, no comparable study is known. The primary claims, i.e., the association of the reported chromosomal loci with heart rate variability measures, are convincing. In this light, further analyses aiming to elucidate a functional role of the detected variants are speculative and of uncertain value (e.g., blood eQTLs rather than cardiomyocyte eQTLs). Thus, the “Potential functional HRV variants” and “Network and functional enrichment analyses” sections as well as the corresponding Discussion sections should be shortened.

REPLY

Given similar concerns of the other reviewers we have trimmed the “Potential functional HRV variants” section removing previous Supplementary tables 16 and 17. We also moved the “Network and functional enrichment analyses” section in its entirety from the main text to the Supplementary Note. The corresponding Discussion sections were trimmed accordingly. With regard to the eQTL (and also mQTL) analyses we beg to differ. We certainly do not contest that large eQTL databases on heart tissue, or ideally SA and AV nodes would certainly have been preferred but such databases are currently small (for heart, see our Koopmans and GTEx analysis in Supplementary Table 15) or non-existent (for SA node). That does not however mean that the identified eQTLs in blood cannot reflect the presence of an eQTL in another more relevant tissue. For many disease traits, including psychiatric disorders^{2,3}, the SNPs identified with GWAS do contain eQTLs in blood even if the primary organ affected is clearly not haematological. In keeping, it has been shown that eQTLs are only partly tissue specific: almost 55% of blood eQTLs are also heart eQTLs for example⁴ (see Figure 2B from this reference below).

Figure 2B from⁴. From the legend: 'Dendrogram and heat map of pairwise eQTL sharing ... $\pi_1 = \Pr(\text{eQTL in tissue } i \text{ given an eQTL in tissue } j)$ '

Our whole blood eQTL examinations in three different independent databases were well powered as these make up the largest eQTL databases in humans. Please note in Supplementary Table 15 that we detect eQTLs for *RGS6*, *HCN* and *GNG11* in the discovery database (NESDA/NTR) and then replicate all three in either a second (BIOS) or third (Westra et al.) blood eQTL database. We also replicate the *GNG11* blood eQTL in two other tissues (tibial artery and medulla).

COMMENT

2. The authors mention the association of heart rate variability with a number of diseases. Here, a heart rate variability loci multi-SNP risk score was only associated with atrial fibrillation but not coronary artery disease, type 2 diabetes... In this reviewer's opinion, this indicates that heart rate variability changes are indeed rather a consequence of the particular diseases. This should be briefly discussed.

REPLY

We fear that absence of the expected associations arose because we only requested the consortia to lookup effects on the disease phenotypes for our 17 genome-wide SNPs for HRV. Although this makes good sense, the variance explained in HRV by these top SNPs is small, and HRV will only be one of many risk factors for these disease outcomes. This approach, therefore, has modest power to detect an association between the HRV SNPs and the disease outcomes, and failure to find the association is not prove of its absence. For the revision we re-contacted the original consortia for these disease risk factors and endpoints with a request to provide us with the full GWAS summary statistics so that we could compute the genetic correlation between HRV and these disease risk factors/endpoints using LD Score regression.

We have added the genetic correlations to the results section with full details in Supplementary Table 11. As you will see the genetic correlations systematically pointed to an overlap in the genetic variants causing low HRV and increased risk for disease (i.e., negative correlations with e.g. SBP, DBP, Coronary Artery Disease, Heart Failure, Sudden Cardiac Death, BMI, and T2D) although significance was reached only for blood pressure after correction for number of outcomes tested. The genetic correlations could of course still be caused by genetic variants for disease as the primary driver, with disease causing low HRV. A strength of this study in this regard is that analyses were confined to individuals in good cardiac health, i.e. cohorts excluded patients with existing cardiovascular diseases or medication potentially impacting HRV. Therefore, our results are compatible with reduced vagal control as a potential risk factor for detrimental cardiovascular outcomes. Even so, the text of our revised discussion still allows for pleiotropy and latent pathology as potential alternatives: *“Using LD Score regression on meta-GWAS summary statistics from various risk factors and endpoints we find some evidence for overlap in the genetic variants causing low HRV and increased risk for disease, but significance was reached only for systolic and diastolic blood pressure after correction for multiple outcomes tested. These genetic correlations are compatible with causal effects of cardiac vagal control in the etiology of disease, but they could also be ascribed to reversed causality, where the disease process leads to lower cardiac vagal control. A strength of this study in this regard is that analyses were confined to individuals in good cardiac health, i.e. cohorts excluded patients with existing cardiovascular diseases or medication potentially impacting HRV. Because we selected individuals in good cardiac health reverse effects of disease on HRV seems less likely, although some latent pathology could have been present. However, an alternative explanation that is harder to rule out is that the genetic correlation derives from pleiotropic effects of genetic variants common to both outcomes.”*

Please note that atrial fibrillation was still positively correlated to low HRV, but we find the correlation unconvincing. LD Score regression results are more telling than just the plots of the non-heterogeneous top SNPs in previous Supplementary Figures 7 and 8. We removed these figures and had to retrace our steps regarding a clear AF – HRV connection after removal of the two heterogeneous SNPs, a connection we now consider too weak to interpret. We adapted the discussion by removing the previous emphasis on AF to reflect this developing wisdom.

COMMENT

3. The most interesting finding in this study to this reviewer is the association of variants in the GNG11, RGS6, and HCN4 genes with heart rate variability. The paper would be strengthened by functional data directly investigating a mechanistic role of the identified SNPs as discussed by the authors (p12, l273-p13, l281).

REPLY

To inspire further experimental work to elucidate the mechanistic role of the identified loci we performed post-GWAS annotation of the HRV SNPs for which we added a flow chart in Supplementary Figure 7 describing the various steps. However, in view of the non-experimental and therefore speculative nature of these analyses we removed them from the main text, adding them to the Supplementary Note.

COMMENT

4. The Discussion is very much focused on the potential pathophysiological influence of the variants. The Discussion would benefit from a focus on rather the genetic findings than speculative implications.

REPLY

We beg to differ of opinion. Much of our discussion is based on Figure 2 that may erroneously have been conceived by the reviewer to have been ‘made up’ based on the current findings. That is not correct. On the contrary, Figure 2 is for the most part based on a very extensive well-established body of knowledge on SA-node signalling pathways– our only addition is the potential effects of the newly identified genetic variants for HRV. We believe these fit and extend this existing body of knowledge in a rather convincing way, although we agree that these hypothesized effects of the HRV variants should be confirmed by experimental evidence, using e.g. knock down variants in valid animal models of HRV.

We take responsibility for inviting this concern from the reviewer by citing only a tiny fragment of the extant literature that supports Figure 2 (so as to comply with the limit placed by the author guidelines on references). We have tried to amend this in the revised version of the legend of the Figure and in the substantially revised version of the discussion.

MINOR COMMENTS

1. Clear written English. Many abbreviations making the paper difficult to read. Careful revision suggested.

REPLY

More of the abbreviations (unless they are used more than 10 times like HRV, or commonly accepted like eQTL) are now spelled out in full. All abbreviations are now defined before they are used and an abbreviation list is added to the Supplementary Information.

2. Clear tables. Figure 1 ok. Figure 2 confusing, speculative, should be revised.

REPLY

Please see our response to your comment 4 above.

Reviewer #3

This manuscript details a large scale, two stage GWAS of three heart rate variability (HRV) traits followed by some in silico functional annotation. HRV is associated with increased risk of sudden cardiac death and all-cause mortality, and has previously been shown to be heritable, suggesting an important genetic component but a large scale GWAS of these traits has not been previously conducted. The authors perform a large number of statistical analyses and find associations in 8 loci, several of which have been previously associated with heart rate and are biologically plausible candidates for regulating HRV, but additional ones that may be independent of HR.

The strengths of the study are in the very large sample size (up to N=53000) with a two-staged discovery and validation and subsequent meta-analysis approach, the somewhat novel phenotype that has been incompletely evaluated at a genetic level, evaluation in non-Caucasian populations, triangulating results to assess prediction of GWAS identified loci with incident events using genotype risk scores, and some in silico functional analysis (with the caveats that that information was not particularly informative). There are however, several issues that temper enthusiasm for the study.

COMMENT

“.. and some in silico functional analysis (with the caveats that that information was not particularly informative)“

REPLY

Given similar concerns of the other reviewers we have removed the “Network and functional enrichment analyses” sections from the main text. It was reorganized, shortened, and placed in the Supplementary Note.

COMMENT

The results are likely only incremental in our understanding of cardiac sinus node regulation as many of the loci have already been shown to be associated with heart rate itself and thus the results likely represent the fact that these variants regulate heart rate and it is known that resting heart rate is correlated with HRV. Thus, the authors should adjust for resting heart rate in all of their analyses to provide a context for true novel findings for HRV vs. those known to affect sinus node regulation through their known effects on resting HR. The authors did do a “look up” in GWAS of heart rate to pull p-values for their HRV loci, but this is insufficient to address this issue.

REPLY

With this comment the reviewer addresses the complexity of the heart rate – HRV relationship, and we begin by admitting that we clearly did not address this important issue in a satisfactory way in the previous version. However, the solution by the reviewer – to correct HRV association analyses for the resting heart rate as a covariate - may also not give sufficient credit to this complex relationship. Indeed, all three measures used by us (SDNN, RMSSD, and pvRSA/HF) are known to be inversely proportional to heart rate. Two parallel mechanisms account for this. The first is unrelated to the underlying biology but simply reflects the definition of HRV. That is,

the slower the heart rate, the longer the inter beat interval, and therefore, any proportionally minor beat-to-beat differences in inter beat interval are more pronounced at slower heart rates. Hence the strong inverse correlation between measures of HRV and heart rate are at least in part due to the manner in which HRV is defined⁵⁻⁷. This alone will cause significant overlap between the genetics of HRV and genetics of heart rate. At the same time, a host of evidence shows that an increase in vagal tone (e.g. by pharmacological means) leads to both decreased heart rate and increased HRV⁸⁻¹³. This fits with current biological understanding of the autonomic regulation of the pacemaker frequency where increased vagal activity, for which HRV is a valid index, is known to causally reduce heart rate.

The complexity that arises in the HRV – heart rate relationship is twofold:

- If that relationship would be dominated by a causal effect of heart rate on HRV, all SNPs for heart rate should have an effect on HRV. They clearly do not (see Supplementary Table 21).
- If that relationship would be dominated by a causal effect of vagal activity on both HRV and heart rate, under the strong assumption that HRV is a perfect index of cardiac vagal activity, all HRV SNPs should influence heart rate. Most do but certainly not all (see Supplementary Table 10). However, we already know that the strong assumption that HRV perfectly indexes cardiac vagal activity is violated, because as we point out in the discussion, HRV also reflects “individual differences in: (i) resting respiration rate and depth¹⁴; (ii) the amplitude of the intrinsic 0.1 Hz oscillations related to both vagal and sympathetic blood pressure regulation through the baroreflex loops¹⁵; (iii) mechanotransduction or intracellular pathways stimulated by sinoatrial stretch¹⁶ or (iv) the efficiency of the actual respiratory-vagal gating process^{17, 18}”.

So, we are faced with the conundrum that HRV is a good measure of vagal activity but not a perfect one. Focusing on SNPs that influence HRV only (but not heart rate) may lead us to biology that is actually unrelated to cardiac vagal activity, our main trait of interest. However, simply correcting HRV for heart rate may “throw out the baby with the bathwater” because a substantial part of the genetic variants for cardiac vagal activity will take the brunt (because they influence heart rate through vagal activity).

With this caveat in mind, the revised version reports on three additional analyses that correct HRV for resting heart rate. First, we used the coefficient of variation of SDNN and RMSSD that corrects for its dependency on the mean heart rate (or more correctly inter beat interval). The coefficient of variation is a measure of spread that describes the amount of inter beat interval variability relative to the mean inter beat interval of each subject, and was recently revived by us to get rid of the proportionality-based dependence of HRV and heart rate¹⁹. Using the coefficients of variation we corrected the meta-analytic results of SDNN and RMSSD with the meta-analytic summary statistics from the resting heart rate meta-analysis, using GWIS²⁰, an analytical approach that can compute the GWAS summary statistics for a mathematical equation-derived variable using the summary statistics of the constituting variables used in the equation. The equations we used to correct HRV for heart rate were

- $(SDNN/X)*100\%$
- $(RMSSD/X)*100\%$

where X is the mean resting inter beat interval which we define as: $X=60000/\text{heart rate}$, because the resting heart rate GWAS meta-analysis used heart rate in beats per

minute, not the inter beat interval. As the coefficients of variation were skewed we used a natural logarithm transformation. Genome-wide significant results of the analyses of these log-transformed coefficients of variation are in the new Supplementary Table 9. Similar analyses could not be performed for pvRSA/HF as effect sizes for pvRSA and HF were not comparable for this combination measure (for this reason we conducted a z-score based meta-analysis for pvRSA/HF).

Secondly, we established the effect of the 17 genome-wide significant SNPs on $\ln((\text{SDNN}/\text{mean inter beat interval}) * 100\%)$ and $\ln((\text{RMSSD}/\text{mean inter beat interval}) * 100\%)$ in Lifelines, NESDA, and TRAILS-Pop, and meta-analyzed the results.

Thirdly, we established the effect of the 17 genome-wide significant SNPs using heart rate as a covariate as suggested by the reviewer in Lifelines, NESDA and TRAILS-Pop, and meta-analyzed the results. As part of this analysis we also calculated the % of the SNP effects on HRV mediated by heart rate and the significance of this mediation using a Sobel test.

We now compared the effect sizes and significance before and after these different corrections for heart rate in the new Supplementary Table 9. The overarching conclusion is that we indeed attenuate some of the HRV SNP effects (in part by overcorrecting a true vagal effect on heart rate and HRV) on average by ~28% (according to the Sobel test of mediation) but the correction for heart rate does not remove the significance of the HRV SNP effects on HRV.

We have added a new section “Correcting HRV for heart rate” to the Results to describe these three additional analyses.

COMMENT

The other major concern is the heterogeneity of the ways HRV was assessed in each cohort including cohorts that included very short tracings. At the minimum, sensitivity analyses of cohorts that have a full 24 hours of tracings to determine HRV should be done.

REPLY

We performed the requested sensitivity analyses for our 17 top SNPs. That is, we conducted analyses on only the 4 cohorts (CHS, FHS, FINGESTURE, ULSAM) using 2-24-hour ambulatory recordings in which posture and activity was not carefully standardized, and compared the results with those from all other cohorts that used laboratory rest recordings. We added these results to a new Supplementary Table 8. We can now report that “*Separately meta-analyzing across cohorts with short laboratory rest recordings versus longer term ambulatory recordings did not suggest sensitivity of the results to these different recording methods (Supplementary Table 8).*”

In addition we point the reviewer to the large correspondence in effect sizes across the various cohorts as shown in the forest plots (Supplementary Information) for our top SNPs and we added the (generally low) I^2 values of the meta-analysis to Supplementary Table 5 confirming that there was little heterogeneity across cohorts.

MINOR COMMENTS

More minor issues are the lack of functional data, particularly for the novel loci

and the large number of analyses that make the manuscript somewhat difficult to follow (adding a figure of study flow with what adjustments for multiple comparisons were made at each stage would be helpful to address this).

REPLY

To inspire further experimental work to elucidate the functional impact of the identified loci we performed post-GWAS annotation of the HRV SNPs for which we added a flow chart in Supplementary Figure 7 describing the various steps. However, in view of the non-experimental and therefore speculative nature of these analyses we removed the gene network and tissue enrichment analyses from the main text, adding them to the Supplementary Note.

We further simplified the analytical approach using headers in the main text to guide the readership through the main steps: *New loci associated with HRV; Variance explained; Generalization to other ethnicities; Association of the HRV variants with resting heart rate; Correcting HRV for heart rate; Association with cardiometabolic traits and diseases; Potential functional impact of the HRV variants*".

Some more minor questions/issues:

1) The authors need to provide justification for their p-value threshold which is not corrected for the number of traits analysed.

REPLY

Because the HRV traits are highly correlated (please see below), Bonferroni adjustment for multiple testing should be considered too strict. Even so, using this correction all genome-wide signals remain significant and we have adjusted the text and footnote to Table 1 accordingly.

2) How correlated were the three HRV traits in their data?

REPLY

An oversight. Possibly induced by the fact that there are many publications addressing the phenotypic correlations between these measures. They seldom reported lower correlations than 0.7²¹⁻²³. For completeness we now report in the Supplementary Note the correlations in three of our larger cohorts covering all possible combinations of pvRSA or HF with SDNN and RMSSD: *"We confirmed the phenotypic correlations in the TRAILS-Pop, NESDA, and Lifelines cohorts where they ranged from 0.70 (SDNN-pvRSA/HF) to 0.96 (RMSSD-pvRSA/HF)."*

3) What were the imputation scores for the top loci?

REPLY

We added the imputation scores (median [interquartile range]) to Supplementary Table 5.

4) Only a small amount of variance in the trait is explained by their loci; although not unexpected the authors should comment on the clinical significance of that.

REPLY

We added this caveat to the discussion: *"Direct clinical relevance of most current GWAS findings is still low and our study is no exception."*

5) Please include OR for prediction of risk for incident events.

REPLY

With the exception of HF, we did not perform lookups in GWAS for incident disease (only prevalent for AF, CAD, SCD, T2D). ORs and betas for continuous outcomes are

reported for the GRSs based on the SNP lookups in Supplementary Table 11. Please note, we now also report in the same table the genetic correlations based on LD score regression of the full GWAS summary statistics of all clinical outcomes.

6) The first paragraph on page 10 is difficult to understand and it is not clear what set of analyses those represent. Was the GSEA done using the RNA microarray data from the subset that had that done on whole blood for HRV traits themselves?

REPLY

We rephrased this paragraph as it clearly led to misunderstanding. Here we did not use RNA microarray data of subjects involved in the cohorts of the HRV meta-analysis. DEPICT works based on existing in silico databases. The corrected text now reads: *“In silico tissue enrichment analysis using DEPICT²⁴ highlights a role for hormones in HRV regulation, with enrichment for the adrenal cortex, endocrine glands, gonads, gastrointestinal tract and female reproductive organs (Supplementary Fig. 9a,c,e). Gene-set enrichment analysis using DEPICT highlights the importance of cardiac development (Supplementary Fig. 9b,d,f).”*

7) Given the linear models used, did the HRV traits meet criteria for parametric analyses after log transformation? This is particularly necessary for the SOLAR analyses that are very sensitive to deviations in normality.

REPLY

In the description of the Oman family study we now made clear that we used a log transformation for all three HRV measures. To allay the reviewers' concern: after log transformation values for skewness and kurtosis for SDNN (skewness = 0.08; kurtosis = 3.81), RMSSD (skewness = -0.34; kurtosis = 3.00) and HF (skewness = 0.05; kurtosis = 3.25) closely resembled those of a normal distribution (skewness = 0; kurtosis = 3.0).

8) Was the genomic inflation rate for each stage?

REPLY

We added the per-cohort genomic control lambda to Supplementary Table 4.

9) Please include a reference for the “double genomic control correction”

REPLY

We added a reference on double genomic control correction²⁵, which was earlier applied in a GWAS meta-analysis for body mass²⁶.

Reference List

1. Notarius, C.F. & Floras, J.S. Caffeine Enhances Heart Rate Variability in Middle-Aged Healthy, But Not Heart Failure Subjects. *J. Caffeine. Res.* **2**, 77-82 (2012).
2. Westra, H.J. *et al.* Systematic identification of trans eQTLs as putative drivers of known disease associations. *Nature Genetics* **1238-1243**, (2013).
3. Schizophrenia Working Group of the Psychiatric Genomics Consortium Biological insights from 108 schizophrenia-associated genetic loci. *Nature* **511**, 421-427 (2014).

4. GTEx Consortium Human Genomics. Human genomics. The Genotype-Tissue Expression (GTEx) pilot analysis: multitissue gene regulation in humans. *Science* **348**, 648-660 (2015).
5. Sacha, J. Interaction between heart rate and heart rate variability. *Ann. Noninvasive. Electrocardiol.* **19**, 207-216 (2014).
6. Sacha, J. Why should one normalize heart rate variability with respect to average heart rate. *Front Physiol* **4**, 306 (2013).
7. Sacha, J. & Pluta, W. Alterations of an average heart rate change heart rate variability due to mathematical reasons. *Int. J. Cardiol.* **128**, 444-447 (2008).
8. Uusitalo, A.L.T., Tahvanainen, K.U.O., Uusitalo, A.J., & Rusko, H.K. Non-invasive evaluation of sympathovagal balance in athletes by time and frequency domain analyses of heart rate and blood pressure variability. *Clin. Physiol.* **16**, 575-588 (1996).
9. Fouad, F.M., Tarazi, R.C., Ferrario, C.M., Fighaly, S., & Alicandri, C. Assessment of parasympathetic control of heart rate by a noninvasive method. *American Journal of Physiology* **246**, H838-H842 (1984).
10. Martinmaki, K., Rusko, H., Kooistra, L., Kettunen, J., & Saalasti, S. Intraindividual validation of heart rate variability indexes to measure vagal effects on hearts. *Am. J. Physiol. Heart. Circ. Physiol.* **290**, H640-H647 (2006).
11. Hayano, J. *et al.* Accuracy of assessment of cardiac vagal tone by heart rate variability in normal subjects. *The American Journal of Cardiology* **67**, 199-204 (1991).
12. Akselrod, S. *et al.* Hemodynamic regulation: investigation by spectral analysis. *American Journal of Physiology* **249**, H867-H875 (1985).
13. Bernston, G.G., Cacioppo, J.T., & Quigley, K.S. Autonomic cardiac control. Estimation and validation from pharmacological blockades. *Psychophysiology* **31**, 572-585 (1994).
14. Grossman, P. & Kollai, M. Respiratory sinus arrhythmia, cardiac vagal tone, and respiration: within- and between-individual relations. *Psychophysiology* **30**, 486-495 (1993).
15. Julien, C. The enigma of Mayer waves: Facts and models. *Cardiovascular Research* **70**, 12-21 (2006).
16. Horner, S.M. *et al.* Contribution to heart rate variability by mechanoelectric feedback. Stretch of the sinoatrial node reduces heart rate variability. *Circulation* **94**, 1762-1767 (1996).
17. Eckberg, D.L. The human respiratory gate. *Journal of Physiology.* **548**, 339-352 (2003).

18. Berntson, G.G., Cacioppo, J.T., & Quigley, K.S. Respiratory sinus arrhythmia: autonomic origins, physiological mechanisms, and psychophysiological implications. *Psychophysiology* **30**, 183-196 (1993).
19. van Roon, A.M., Snieder, H., Lefrandt, J.D., de Geus, E.J., & Riese, H. Parsimonious Correction of Heart Rate Variability for Its Dependency on Heart Rate. *Hypertension* **68**, e63-e65 (2016).
20. Nieuwboer, H.A., Pool, R., Dolan, C.V., Boomsma, D.I., & Nivard, M.G. GWIS: Genome-Wide Inferred Statistics for Functions of Multiple Phenotypes. *Am. J. Hum. Genet.* **99**, 917-927 (2016).
21. Goedhart, A.D., van der Sluis, S., Houtveen, J.H., Willemsen, G., & de Geus, E.J. Comparison of time and frequency domain measures of RSA in ambulatory recordings. *Psychophysiology* **44**, 203-215 (2007).
22. Kupper, N.H. *et al.* Heritability of ambulatory heart rate variability. *Circulation* **110**, 2792-2796 (2004).
23. Grossman, P., van Beek, J., & Wientjes, C. A comparison of three quantification methods for estimation of respiratory sinus arrhythmia. *Psychophysiology* **27**, 702-714 (1990).
24. Pers, T.H. *et al.* Biological interpretation of genome-wide association studies using predicted gene functions. *Nature Communications* **6**, 5890 (2015).
25. Devlin, B. & Roeder, K. Genomic control for association studies. *Biometrics* **55**, 997-1004 (1999).
26. Lindgren, C.M. *et al.* Genome-wide association scan meta-analysis identifies three Loci influencing adiposity and fat distribution. *PLoS Genetics* **5**, e1000508 (2009).

Reviewer #1 (Remarks to the Author):

Thank you for addressing the comments and updating the draft accordingly. I just have a couple minor comments.

Lines 15,16 in abstract gets repeated in lines 54,55 and by the time one reaches the end of introduction, there is no data/result to back up this statement.

Lines 248-250: Not sure if I understand the phrase "sort their effect"?

Reviewer #2 (Remarks to the Author):

The authors have adressed my concerns. No further commen

REVIEWERS' COMMENTS:

Reviewer #1 (Remarks to the Author):

Thank you for addressing the comments and updating the draft accordingly. I just have a couple minor comments.

COMMENT

Lines 15,16 in abstract gets repeated in lines 54,55 and by the time one reaches the end of introduction, there is no data/result to back up this statement.

REPLY

Lines 15.16: In the abstract we now refer back to the found genes so as to connect this statement with the reported results, i.e. the sentence was changed to ‘with a key role for genetic variants (*GNG11*, *RGS6*) that influence G-protein heterotrimer action in GIRK- channel induced pacemaker membrane hyperpolarization.’

Lines 54,55 : If the reviewer means that we provide a summary of the main results at the end of the introduction before full disclosure of the actual results then we must point out this was per the request of Nature Communications to add such a summary. If the reviewer means that the link to the GIRK-channel induced pacemaker membrane hyperpolarization comes a bit out of the blue, we can agree. Again we solve this by referring back to the found genes (*GNG11*, *RGS6*)”for two proteins well-known to regulate GIRK-channel induced pacemaker membrane hyperpolarization”.

COMMENT

Lines 248-250: Not sure if I understand the phrase "sort their effect"?

REPLY

We have rephrased to: ‘exert their effect’.

No other comments remained.